# The genetic landscape for amyloid beta fibril nucleation accurately discriminates familial Alzheimer's disease mutations

**Mireia Seuma[1], Andre J Faure[2], Marta Badia[1], Ben Lehner[2,3,4]\*, Benedetta Bolognesi[1]\***

[1]Institute for Bioengineering of Catalonia (IBEC), The Barcelona Institute of Science and Technology, Barcelona, Spain; [2]Center for Genomic Regulation (CRG), The Barcelona Institute of Science and Technology, Barcelona, Spain; [3]Universitat Pompeu Fabra (UPF), Barcelona, Spain; [4]ICREA, Pg. Lluís Companys, Barcelona, Spain

**Abstract** Plaques of the amyloid beta (Aß) peptide are a pathological hallmark of Alzheimer's disease (AD), the most common form of dementia. Mutations in Aß also cause familial forms of AD (fAD). Here, we use deep mutational scanning to quantify the effects of >14,000 mutations on the aggregation of Aß. The resulting genetic landscape reveals mechanistic insights into fibril nucleation, including the importance of charge and gatekeeper residues in the disordered region outside of the amyloid core in preventing nucleation. Strikingly, unlike computational predictors and previous measurements, the empirical nucleation scores accurately identify all known dominant fAD mutations in Aß, genetically validating that the mechanism of nucleation in a cell-based assay is likely to be very similar to the mechanism that causes the human disease. These results provide the first comprehensive atlas of how mutations alter the formation of any amyloid fibril and a resource for the interpretation of genetic variation in Aß.

**\*For correspondence:**
ben.lehner@crg.eu (BL);
bbolognesi@ibecbarcelona.eu (BB)

**Competing interests:** The authors declare that no competing interests exist.

## Introduction

Amyloid plaques consisting of the amyloid beta (Aß) peptide are a pathological hallmark of Alzheimer's disease (AD), the most common cause of dementia and a leading global cause of morbidity with very high societal and economic impact (*Ballard et al., 2011*; *World Health Organization, 2012*). Although most cases of AD are sporadic and of uncertain cause, rare familial forms of the disease also exist (*Campion et al., 1999*). These inherited forms of dementia typically have earlier onset and are caused by high penetrance mutations in multiple loci, including in the amyloid precursor protein (*APP*) gene, which encodes the protein from which Aß is derived by proteolytic cleavage (*O'Brien and Wong, 2011*). Several mutations in *PSEN1* and *PSEN2*, the genes coding for the secretases performing sequential cleavage of APP, also lead to autosomal dominant forms of AD. The two most abundant isoforms of Aß generated upon cleavage are 42 and 40 amino acids (aa) in length, with the longer Aß peptide associated with increased aggregation in vitro and cellular toxicity (*Meisl et al., 2014*; *Sandberg et al., 2010*). The amyloid state is a thermodynamically low energy state but, both in vitro and in vivo, the spontaneous formation of amyloids is normally very slow because of the high kinetic barrier of fibril nucleation (*Knowles et al., 2014*). The process of nucleation generates oligomeric Aß species that have been hypothesized to be particularly toxic to cells and that then grow into fibrils (*Michaels et al., 2020*; *Bolognesi et al., 2010*; *Cleary et al., 2005*).

Fourteen different mutations in the Aß peptide have been reported to cause familial Alzheimer's disease (fAD), with all but two having a dominant pattern of inheritance (*Weggen and Beher, 2012*; *Van Cauwenberghe et al., 2016*). However, it is not clear why these particular mutations cause fAD

**eLife digest** Alzheimer's disease is the most common form of dementia, affecting more than 50 million people worldwide. Despite more than 400 clinical trials, there are still no effective drugs that can prevent or treat the disease. A common target in Alzheimer's disease trials is a small protein called amyloid beta. Amyloid beta proteins are 'sticky' molecules. In the brains of people with Alzheimer's disease, they join to form first small aggregates and then long chains called fibrils, a process which is toxic to neurons.

Specific mutations in the gene for amyloid beta are known to cause rare, aggressive forms of Alzheimer's disease that typically affect people in their fifties or sixties. But these are not the only mutations that can occur in amyloid beta. In principle, any part of the protein could undergo mutation. And given the size of the human population, it is likely that each of these mutations exists in someone alive today.

Seuma et al. reasoned that studying these mutations could help us understand the process by which amyloid beta forms new aggregates. Using an approach called deep mutational scanning, Seuma et al. mutated each point in the protein, one at a time. This produced more than 14,000 different versions of amyloid beta. Seuma et al. then measured how quickly these mutants were able to form aggregates by introducing them into yeast cells.

All the mutations known to cause early-onset Alzheimer's disease accelerated amyloid beta aggregation in the yeast. But the results also revealed previously unknown properties that control how fast aggregation occurs. In addition, they highlighted a number of positions in the amyloid beta sequence that act as 'gatekeepers'. In healthy brains, these gatekeepers prevent amyloid beta proteins from sticking together. When mutated, they drive the protein to form aggregates.

This comprehensive dataset will help researchers understand how proteins form toxic aggregates, which could in turn help them find ways to prevent this from happening. By providing an 'atlas' of all possible amyloid beta mutations, the dataset will also help clinicians interpret any new mutations they encounter in patients. By showing whether or not a mutation speeds up aggregation, the atlas will help clinicians predict whether that mutation increases the risk of Alzheimer's disease.

(*Weggen and Beher, 2012*; *Van Cauwenberghe et al., 2016*), and these 14 mutations represent only 3.7% of the possible 378 single nucleotide changes that can occur in Aß. As for nearly all disease genes, therefore, the molecular mechanism by which mutations cause the disease remains unclear and the vast majority of possible mutations in Aß are variants of uncertain significance (VUS). This makes the clinical interpretation of genetic variation in this locus a difficult challenge (*Starita et al., 2017*; *Gelman et al., 2019*). Moreover, given the human mutation rate and population size, it is likely that nearly all of these possible variants in Aß actually exist in at least one individual currently alive on the planet (*Conrad et al., 2011*). A comprehensive map of how all possible mutations affect the formation of Aß amyloids and how these changes relate to the human disease is therefore urgently needed.

More generally, amyloid fibrils are associated with many different human diseases (*Knowles et al., 2014*), but how mutations alter the propensity of proteins to aggregate into amyloid fibrils is not well understood and there has been no large-scale analysis of the effects of mutations on the formation of any amyloid fibril. Here, we address this shortcoming by quantifying the rate of fibril formation for >14,000 variants of Aß. This provides the first comprehensive map of how mutations alter the propensity of any protein to form amyloid fibrils. The resulting data provide numerous mechanistic insights into the process of Aß fibril nucleation. Moreover, they also accurately classify all the known dominant fAD mutations, validating the clinical relevance of a simple cell-based model and providing a comprehensive resource for the interpretation of clinical genetic data.

## Results

### Deep mutagenesis of Aß

To globally quantify the impact of mutations on the nucleation of Aß fibrils, we used an in vivo selection assay in which the nucleation of Aß is rate-limiting for the aggregation of a second amyloid, the yeast prion [*PSI+*] encoded by the *sup35* gene (*Chandramowlishwaran et al., 2018*). Aggregation of Sup35p causes read-through of UGA stop codons, allowing growth-based selection using an auxotrophic marker containing a premature termination codon (*Figure 1A* and *Figure 1—figure supplement 1A*). We generated a library containing all possible single nucleotide variants of Aß42 fused to the nucleation (N) domain of Sup35p and quantified the effect of mutations on the rate of nucleation in triplicate by selection and deep sequencing (*Faure et al., 2020*; see Materials and methods). The selection assay was highly reproducible, with enrichment scores for aa substitutions strongly correlated between replicates (*Figure 1B* and *Figure 1—figure supplement 1B*).

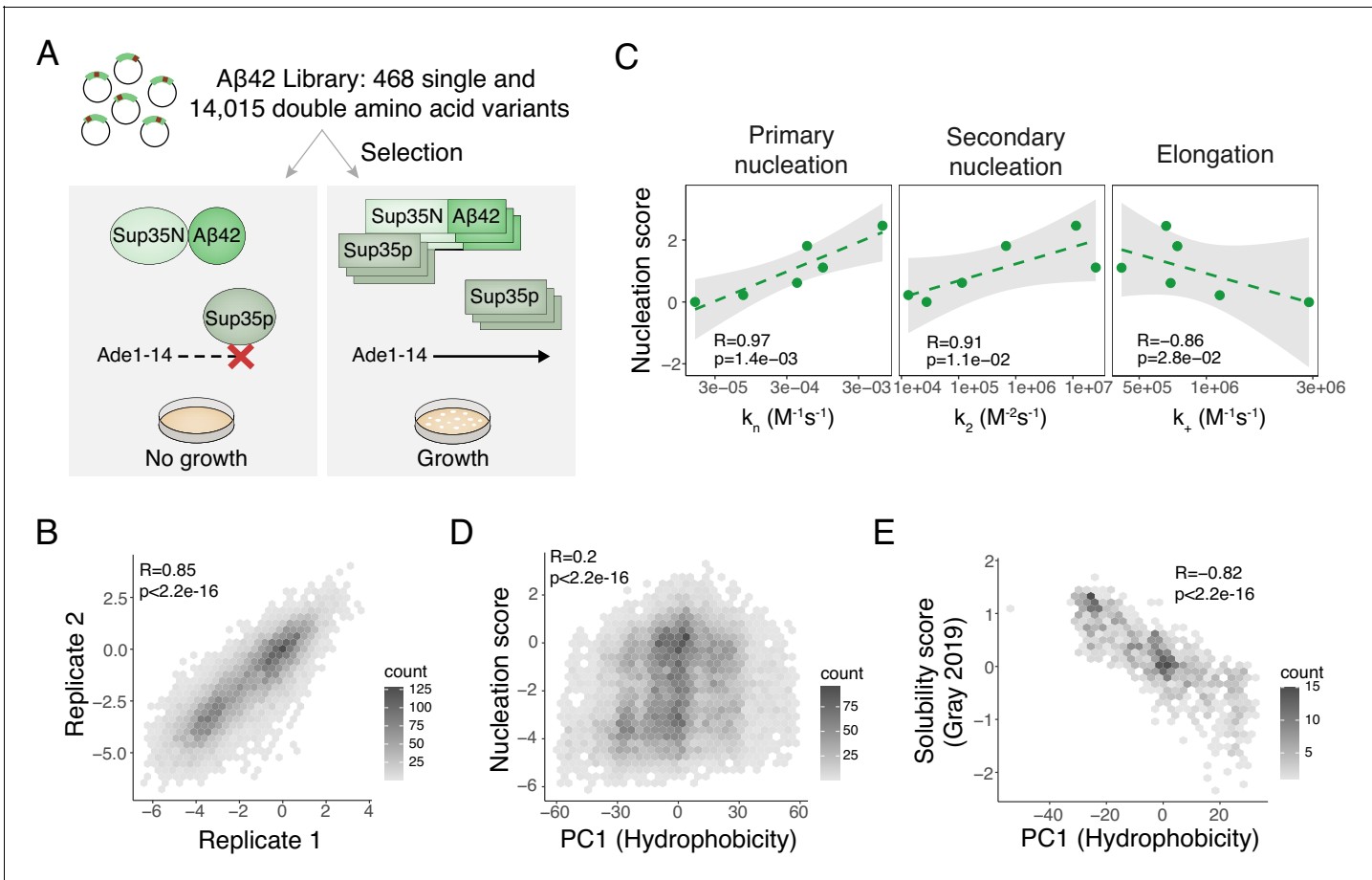

**Figure 1.** Deep mutagenesis of amyloid beta (Aß) nucleation. (**A**) In vivo Aß selection assay. Aß fused to the Sup35N domain seeds aggregation of endogenous Sup35p causing a read-through of a premature UGA in the *Ade1-14* reporter, allowing the cells to grow in medium lacking adenine. (**B**) Correlation of nucleation scores for biological replicates 1 and 2 for single and double amino acid (aa) mutants. Pearson correlation coefficient and p-value are indicated (*Figure 1—figure supplement 1B*) n = 10,157 genotypes. (**C**) Correlation of nucleation scores with in vitro primary and secondary nucleation and elongation rate constants (*Yang et al., 2018*). Weighted Pearson correlation coefficient and p-value are indicated. (**D**) Nucleation scores as a function of principal component 1 (PC1) aa property changes (*Bolognesi et al., 2019*) for single and double aa mutants (n = 14,483 genotypes). Weighted Pearson correlation coefficient and p-value are indicated. (**E**) Solubility scores (*Gray et al., 2019*) as a function of PC1 changes (*Bolognesi et al., 2019*) for n = 895 single and double mutants. Pearson correlation coefficient and p-value are indicated.

The online version of this article includes the following source data and figure supplement(s) for figure 1:

**Figure supplement 1.** Reproducibility of the assay and correlation with in vitro fibril nucleation.

**Figure supplement 1—source data 1.** Raw colony counts from independent testing of the strains expressing the variants reported in *Figure 1—figure supplement 1A*.

### In vivo nucleation scores are highly correlated with in vitro rates of amyloid nucleation

Comparing our in vivo enrichment scores to the qualitative effects of 16 mutations analysed in vitro across 10 previous publications validated the assay, with mutational effects matching the effects on in vitro nucleation previously reported for 14 Aß variants out of 16. (*Supplementary file 1*). Moreover, the in vivo scores correlate extremely well with the rate of nucleation of Aß variants in positions 21, 22, 23 (*Yang et al., 2018*; *Törnquist et al., 2018*; *Figure 1C* and *Figure 1—figure supplement 1C*). We henceforth refer to the in vivo enrichment scores as 'nucleation scores' (NS).

### Two mechanisms of in vivo Aß aggregation

A prior deep mutational scan quantified the effects of mutations on the abundance of Aß fused to an enzymatic reporter (*Gray et al., 2019*). These 'solubility scores' do not predict the effects of mutations on Aß nucleation (*Figure 1—figure supplement 1D*). Previously we identified a principal component of aa properties (principal component 1 [PC1], related to changes in hydrophobicity) that predicts the aggregation and toxicity of the amyotrophic lateral sclerosis (ALS) protein TDP-43 when it is expressed in yeast (*Bolognesi et al., 2019*). PC1 is also not a good predictor of Aß nucleation (*Figure 1D*) but it does predict the previously reported changes in Aß solubility (*Figure 1E*), suggesting that Aß is aggregating by a similar process to TDP-43 in this alternative selection assay (*Gray et al., 2019*) but by a different mechanism in the nucleation selection.

### Nucleation scores for 14,483 Aß variants

The distribution of mutational effects for Aß nucleation has a strong bias towards reduced nucleation, with 56% of single aa substitutions reducing nucleation but only 16% increasing it (Z-test, false discovery rate [FDR] = 0.1, *Figure 2A*). Moreover, mutations that decrease nucleation in our dataset typically have a larger effect than those that increase it, with many mutations reducing nucleation to the background rate observed for Aß variants containing premature termination codons (*Figure 2A*).

In addition to covering all aa changes obtainable through single nt mutations, our mutagenesis library was designed to contain a substantial fraction of double mutants. In total, we quantified the impact of 14,015 double aa variants of Aß. Double mutants were even more likely to reduce nucleation, with 63% decreasing and only 5.5% increasing nucleation (Z-test, FDR = 0.1; *Figure 2B*). Therefore, mutations more frequently decrease rather than increase Aß nucleation.

### Aß has a modular mutational landscape

Inspecting the heatmap of mutational effects for aa changes at all positions in Aß reveals strong biases in the locations of mutations that increase and decrease nucleation (*Figure 2C and D*, and *Figure 2—figure supplement 1A*). Mutations that decrease nucleation are highly enriched in the C-terminus of Aß, whereas mutations that increase nucleation are enriched in the N-terminus (*Figure 2E*). Indeed, >84% of mutations in the C-terminus (residues 27-42) reduce nucleation and only 9.6% increase it (FDR = 0.1). In contrast, the effects of mutations are smaller (*Figure 2F*) and also more balanced in the first 26 aa of the peptide, with 38.6% decreasing and 20% increasing nucleation (FDR = 0.1).

These differences in the direction and strength of mutational effects between the N- and C-terminal regions of Aß suggest a modular organization of the peptide. This modularity is also reflected in the primary sequence of Aß, which has a hydrophobic C-terminus and a more polar and charged N-terminus (eight out of nine charged residues in Aß are found before residue 24 and the peptide consists entirely of hydrophobic residues from position 29) (*Figures 2C* and *3A*). Consistent with this modular organization, mutations in the few hydrophobic residues in the N-terminus have effects that are more similar to mutations in polar residues in the N-terminus rather than in hydrophobic residues in the C-terminus. Similarly, mutations in the most C-terminal charged residue (K28) frequently strongly reduce nucleation, just as they do in the adjacent hydrophobic positions (*Figure 3A*).

### Gatekeeper residues act as anti-nucleators

Considering the entire Aß peptide, there are only seven positions in which mutations are not more likely to decrease rather than increase nucleation (FDR = 0.1; *Figure 2D*). Strikingly, these positions,

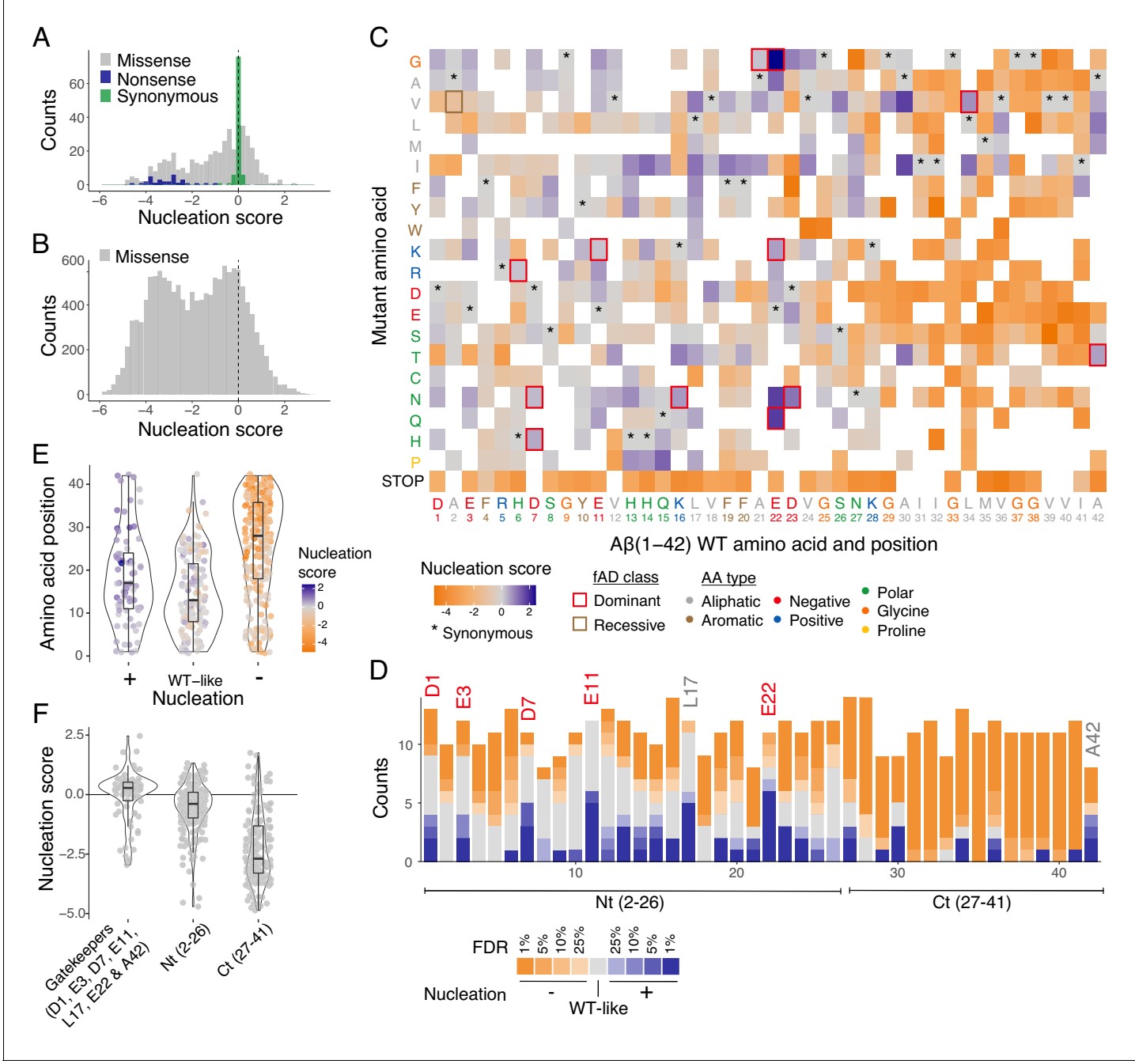

**Figure 2.** Modular organization of mutational effects in amyloid beta (Aß). (**A and B**) Nucleation scores distribution for single (**A**) and double (**B**) amino acid (aa) mutants. n = 468 (missense), n = 31 (nonsense), n = 90 (synonymous) for singles, and n = 14,015 (missense) for doubles. Vertical dashed line indicates wild-type (WT) score (0). (**C**) Heatmap of nucleation scores for single aa mutants. The WT aa and position are indicated in the x-axis and the mutant aa is indicated on the y-axis, both coloured by aa class. Variants not present in the library are represented in white. Synonymous mutants are indicated with '*' and familial Alzheimer's disease (fAD) mutants with a box, coloured by fAD class. (**D**) Number of variants significantly increasing (blue) and decreasing (orange) nucleation at different false discovery rates (FDRs). Gatekeeper positions (D1, E3, D7, E11, L17, E22, and A42) are indicated on top of the corresponding bar and coloured on the basis of aa type. The N-terminal and C-terminal definitions are indicated on the x-axis. Gatekeeper positions are excluded from the N-terminal and C-terminal classes. (**E**) Aa position distributions for variants that increase (+), decrease (−), or have no effect on nucleation (WT-like) (FDR < 0.1). (**F**) Nucleation score distributions for the three clusters of positions defined on the basis of nucleation: Nt (2-26), Ct (27-41), and gatekeeper positions (clusters are mutually exclusive). Horizontal line indicates WT nucleation score (0). Nonsense (stop) mutants were only included in **A** and **C**. Boxplots represent median values and the lower and upper hinges correspond to the 25th and 75th percentiles, respectively. Whiskers extend from the hinge to the largest value no further than 1.5*IQR (interquartile range). Outliers are plotted individually or omitted when the boxplot is plotted together with individual data points or a violin plot.

*Figure 2 continued on next page*

*Figure 2 continued*

The online version of this article includes the following figure supplement(s) for figure 2:

**Figure supplement 1.** Mutational effects in amyloid beta (Aß).

which we refer to as 'gatekeepers' of nucleation (*Rousseau et al., 2006*; *Pedersen et al., 2004*), include five of the six negatively charged residues in Aß. The sixth gatekeeper is an unusual hydrophobic residue in the N-terminus, L17, where seven mutations increase nucleation and only one decreases it (FDR = 0.1; *Figure 2D*). The final aa of the peptide, A42, also has an unusual distribution of mutational effects that is different to the rest of the C-terminus, with four mutations increasing and three mutations decreasing nucleation (FDR = 0.1; *Figure 2D*).

Taken together, on the basis of mutational effects, we therefore distinguish the following mutually exclusive positions in Aß: the C-terminus (aa 27-41) where the majority of mutations strongly decrease nucleation, the N-terminus (aa 2-26) where mutations have smaller and more balanced effects, and seven gatekeeper residues (D1, E3, D7, E11, D22, L17, A42) where mutations frequently increase nucleation. We consider each of these classes below.

## Mutations in the N- and C-terminal regions

Mutations in the C-terminus nearly all decrease nucleation (*Figure 3A*). This is consistent with the C-terminus forming part of the tightly packed amyloid core of all known structural polymorphs of both Aß42 (*Colvin et al., 2016*; *Meier et al., 2017*; *Wälti et al., 2016*; *Xiao et al., 2015*; *Gremer et al., 2017*; *Lührs et al., 2005*; *Schmidt et al., 2015*) and Aß40 (*Kollmer et al., 2019*; *Lu et al., 2013*; *Qiang et al., 2012*; *Sgourakis et al., 2015*; *Paravastu et al., 2008*; *Schütz et al., 2015*). Consistent with this, we quantified the nucleation of three C-terminal fragments of the peptide (aa 22-42, 24-42, 27-42) and found that they nucleate similarly or better than full length Aß (*Figure 3—figure supplement 1C*). Mutations to polar and charged residues in this region nearly all decrease nucleation, but so too do most changes to other hydrophobic residues (*Figure 3B*), suggesting specific side chain packing in this region is important for nucleation. The relative effects of different mutations are only partially captured by changes in hydrophobicity (*Figure 3F*; Pearson correlation coefficient, R = 0.45) and by predictors of aggregation potential (*Figure 3—figure supplement 1A*). Only a few mutations in this region increase nucleation: substitutions to isoleucine at positions 30, 34, and 39; mutations to valine at positions 29, 30, and 34; a change to threonine at position 30; changes to leucine and methionine at 36; and a mutation to phenylalanine at position 41 (FDR = 0.1).

Mutations in the N-terminus of Aß have a more balanced effect on nucleation, and these effects are not well predicted by either hydrophobicity or predictors of aggregation potential (*Figure 3—figure supplement 1B,D and E*). The effects of introducing particular aa are, however, biased, with the introduction of asparagine, isoleucine, and valine most likely to increase nucleation (*Figure 3C* and *Figure 3—figure supplement 2*). As at the C-terminus, the introduction of negative charged residues typically strongly reduces nucleation (*Figure 3B and C*). However, in contrast to what is observed in the C-terminus (*Figure 3B*), the effects of introducing positive charge are less severe (*Figure 3C*). Interestingly, the effects of mutations to proline, isoleucine, valine, and threonine in the N-terminus depend on the position in which they are made: mutations in the first 12 residues typically decrease nucleation, whereas mutations in the next four to nine residues increase nucleation (*Figure 3E*). The conformational rigidity of proline and the beta-branched side chains of isoleucine, valine, and threonine that disfavour helix formation suggest that disruption of a secondary structure in this region may favour nucleation. Interestingly, this same region was highlighted as the part of the peptide remaining most disordered across different states of the solution ensemble of Aß in molecular dynamics simulations, with the same region also making extensive long-range contacts in different states of the kinetic ensemble (*Löhr et al., 2021*).

## The role of charge in limiting Aß nucleation

At five of six negatively charged positions in Aß, mutations frequently increase nucleation (*Figures 2D* and *3A*). Moreover, the introduction of negative charge at other positions strongly decreases nucleation (*Figure 3A*), suggesting that negatively charged residues act as gatekeepers

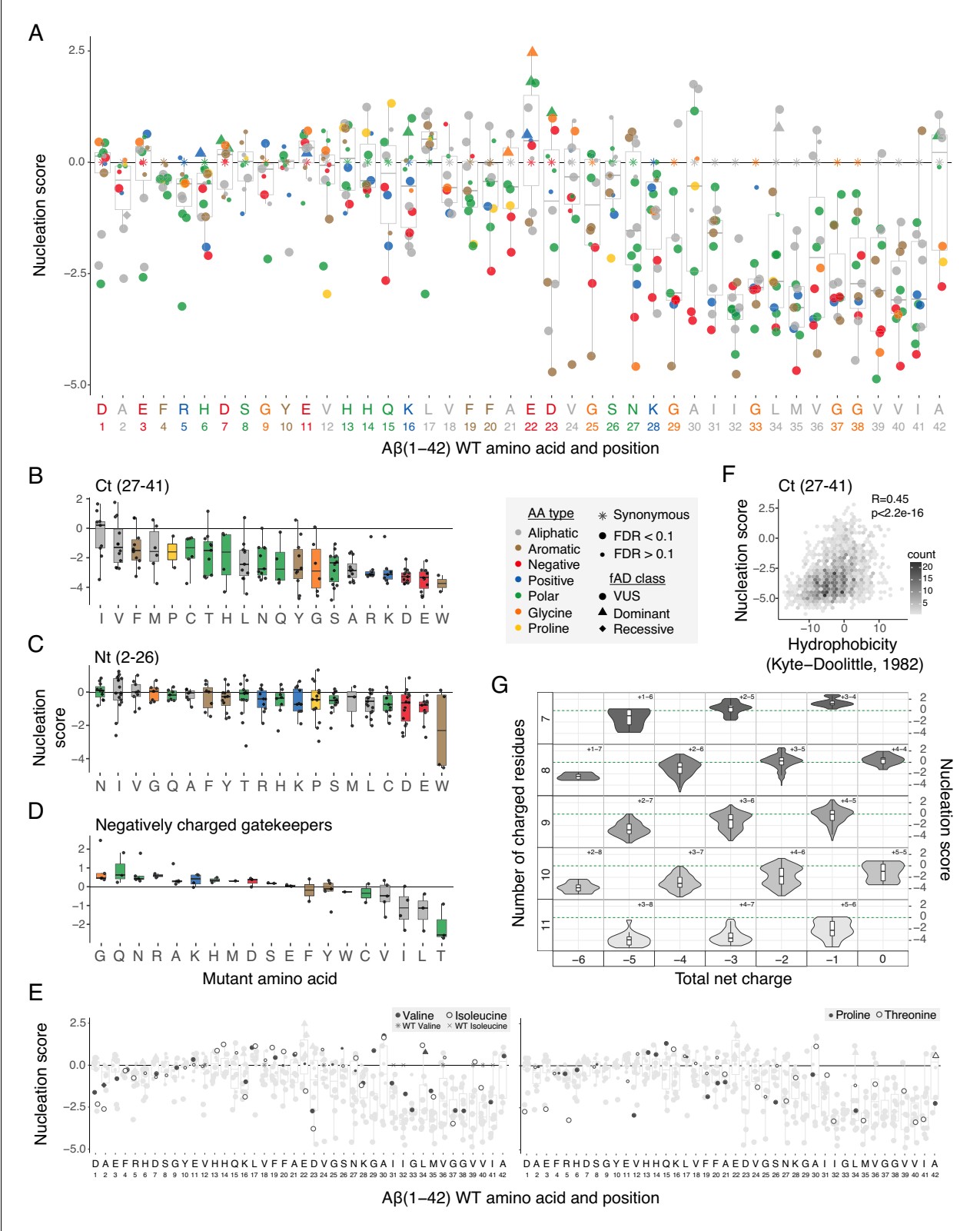

**Figure 3.** Determinants of amyloid beta (Aß) nucleation. (**A**) Effect of single aa mutants on nucleation for each Aß position. The wild-type (WT) aa and position are indicated on the x-axis and coloured on the basis of aa type. The horizontal line indicates the WT nucleation score (0). (**B to D**) Effect of each mutant aa on nucleation for the Ct (27-41) (**B**), the Nt (2-26) (**C**), and the negatively charged gatekeeper positions (D1, E3, D7, E11, and E22) (**D**). The three position clusters are mutually exclusive. Colour code indicates aa type. The horizontal line is set at the WT nucleation score (0). (**E**) Effect on

*Figure 3 continued on next page*

*Figure 3 continued*

nucleation for single aa mutations to proline, threonine, valine, and isoleucine. Mutations to other aa are indicated in grey. The horizontal line indicates WT nucleation score (0). Point size and shape indicate false discovery rate (FDR) and familial Alzheimer's disease (fAD) class, respectively (see legend). (F) Nucleation scores as a function of hydrophobicity changes (*Kyte and Doolittle, 1982*) for single and double aa mutants in the Ct (27-41) cluster. Only double mutants with both mutations in the indicated position-range were used. Weighted Pearson correlation coefficient and p-value are indicated. (G) Nucleation score distributions arranged by the number of charged residues (y-axis) and the total net charge (x-axis) for single and double aa mutants in the full peptide (1-42). Only polar, charged, and glycine aa types were taken into account, for both WT and mutant residues. Colour gradient indicates the total number of charged residues. Numbers inside each cell indicate the number of positive and negative residues. The horizontal line indicates WT nucleation score (0). Boxplots represent median values and the lower and upper hinges correspond to the 25th and 75th percentiles, respectively. Whiskers extend from the hinge to the largest value no further than 1.5*IQR (interquartile range). Outliers are plotted individually or omitted when the boxplot is plotted together with individual data points or a violin plot.

The online version of this article includes the following source data and figure supplement(s) for figure 3:

**Figure supplement 1.** Determinants of amyloid beta (Aß) nucleation.

**Figure supplement 1—source data 1.** Raw colony counts from indepednet testing of the strains expressing the N-terminal truncated varaints reported in *Figure 3—figure supplement 1C*.

**Figure supplement 2.** Effect of mutations to each specific amino acid (aa) on amyloid beta (Aß) nucleation.

---

(*Pedersen et al., 2004*; *Rousseau et al., 2006*) to limit nucleation (*Figure 3D* and *Figure 3—figure supplement 1D*). In contrast, mutations in the three positively charged residues (R5, K16, K28) mostly decrease nucleation (*Figure 2D*). Mutating the negatively charged gatekeepers to the polar aa glutamine and asparagine, to positively charged residues (arginine and lysine), or to small side chains (glycine and alanine) increases nucleation (*Figure 3D*). Mutating the same positions to hydrophobic residues typically reduces nucleation (*Figure 3D*). This is consistent with a model in which the negative charge at these positions acts to limit nucleation, but that the overall polar and unstructured nature of the N-terminus must be maintained for effective nucleation.

To further investigate the role of charge in controlling Aß nucleation, we extended our analyses to the double mutants. Including double mutants allows the net charge of Aß to vary over a wider range and it also allows comparison of the nucleation of peptides with the same net charge but a different total number of charged residues (e.g., a net charge of −3 can result from a negative/positive aa composition of 6/3, as in wild-type Aß, or compositions of 7/4, 5/2, etc.). Considering all mutations between charged and polar residues or glycine reveals that, although reducing the net charge of the peptide from −3 progressively increases nucleation (*Figure 3G*), the total number of charged residues is also important: for a given net charge, nucleation is increased in peptides containing fewer charged residues of any sign (*Figure 3G* and *Figure 3—figure supplement 1F and G*). Thus, both the overall charge and the number of charged residues control the rate of Aß nucleation.

## Hydrophobic gatekeeper residues

In addition to the five negatively charged gatekeeper residues, mutations most frequently increase nucleation of Aß in two specific hydrophobic residues: L17 and A42 (*Figure 2C and D*). At position 17, changes to polar, aromatic, and aliphatic aa all increase nucleation, as does the introduction of a positive charge and mutation to proline. Only a mutation to cysteine reduces nucleation (*Figure 2C*). This suggests a specific role for leucine at position 17 in limiting nucleation, perhaps as part of a nucleation-limiting secondary structure suggested by the mutational effects of proline, isoleucine, valine, and threonine in this region (*Figure 3E*).

Finally, the distribution of mutational effects at position 42 differs from that in the rest of the hydrophobic C-terminus of Aß, with mutations most often increasing nucleation (*Figure 2D*; FDR = 0.1). The mutations that increase nucleation are all to other aliphatic residues (*Figures 2C and 3A*). The distinction of position 42 is interesting because of the increased toxicity and aggregation propensity of Aß42 compared to the shorter Aß40 APP cleavage product (*Meisl et al., 2014*; *Sandberg et al., 2010*).

## Nucleation scores accurately discriminate fAD mutations

To investigate how nucleation in the cell-based assay relates to the human disease, we considered all the mutations in Aß known to cause fAD. In total, there are 11 mutations in Aß reported to cause dominantly inherited fAD and one additional variant of unclear pathogenicity (H6R) (*Janssen et al.,*

*2003*). These 12 known disease mutations are not well discriminated by commonly used computational variant effect predictors (*Figure 4* and *Figure 4—figure supplement 1A*) or by computational predictors of protein aggregation and solubility (*Figure 4* and *Figure 4—figure supplement 1B*). They are also poorly predicted by the previous deep mutational scan of Aß designed to quantify changes in protein solubility, suggesting the disease is unrelated to the biophysical process quantified in this assay (*Gray et al., 2019*; *Figure 4—figure supplement 1C*).

In contrast, the scores from our in vivo nucleation assay accurately classify the known dominant fAD mutations, with all 12 mutations increasing nucleation (*Figure 4*, area under the receiver operating characteristic curve, ROC−AUC = 0.9, two-tailed Z-test, p<2.2e-16). This suggests the biophysical events occurring in this simple cell-based assay are highly relevant to the development of the human disease.

Consistent with the overall mutational landscape, the known fAD mutations are also enriched in the N-terminus of Aß (*Figure 2C*). In some positions the known fAD mutations are the only mutation or one of only a few mutations that can increase nucleation. For example, based on our data, K16N is likely to be one of only two fAD mutations in position 16. However, in other positions, there are several additional variants that increase nucleation as much as the known fAD mutation. At position 11, for example, there are five mutations with a NS higher than the known E11K disease mutation (*Figure 2C and D*). Overall, our data suggest there are likely to be many additional dominant fAD mutations beyond the 12 that have been reported to date (*Supplementary file 2*).

In addition to the 12 known dominant fAD mutations, two additional variants in Aß have been suggested to act recessively to cause fAD (*Di Fede et al., 2009*; *Tomiyama et al., 2008*). One of these variants is a codon deletion (E22Δ) and is not present in our library. The other variant, A2V, does not have a dominant effect on nucleation in our assay (*Figure 2C*), consistent with a recessive pattern of inheritance and a different mechanism of action, such as reduced ß-cleavage and increased Aß42 generation, as previously proposed (*Benilova et al., 2014*). More generally, of the hundreds of aa changes possible in the peptide, our data prioritize 63 as candidate fAD variants (*Supplementary file 2*); 131 variants are likely to be benign, and 262 reduce Aß nucleation and so may even be protective. These include variants already reported in the gnomAD database of human genetic variation (*Figure 4—figure supplement 1D*). With the currently available data for patients

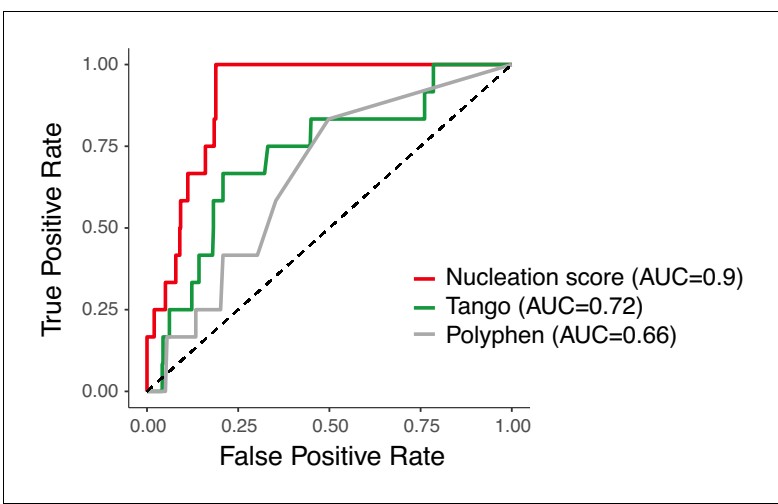

**Figure 4.** Amyloid beta (Aß) nucleation accurately discriminates dominant familial Alzheimer's disease (fAD) variants. Receiver operating characteristic (ROC) curves for 12 fAD mutants versus all other single aa mutants in the dataset. Area under the curve (AUC) values are indicated in the legend. Diagonal dashed line indicates the performance of a random classifier. The nucleation scores and categories for all fAD variants are reported in *Supplementary file 1*.

The online version of this article includes the following figure supplement(s) for figure 4:

**Figure supplement 1.** Discrimination of familial Alzheimer's disease (fAD) variants by aggregation and variant effect predictors.

carrying fAD mutations, we could not observe a correlation between NS and disease age-of-onset (*Ryman et al., 2014*; *Figure 4—figure supplement 1E*).

## Discussion

Taken together, the data presented here provides the first large-scale analysis of how mutations promote and prevent the aggregation of an amyloid. The results reveal a modular organization for the impact of mutations on the nucleation of Aß. Moreover, they show that the rate of nucleation in a cell-based assay identifies all of the mutations in Aß that cause dominant fAD. The dataset therefore provides a useful resource for the future clinical interpretation of genetic variation in Aß.

A majority of mutations in the C-terminal core of Aß disrupt nucleation, consistent with specific hydrophobic contacts in this region being required for nucleation. In contrast, mutations that increase nucleation are enriched in the polar N-terminus with mutations in negatively charged gatekeeper residues and the L17 gatekeeper being particularly likely to accelerate aggregation. Indeed, decreasing both the net charge of the peptide and the total number of charged residues increases nucleation.

Little is known about the structure of Aß during fibril nucleation, but the results presented here are in general consistent with the nucleation transition state resembling the known mature fibril structures of Aß where the C-terminal region of the peptide is located in the amyloid core and the N-terminus is disordered and solvent exposed (*Figure 5* and *Figure 5—figure supplements 1* and *2*). Although the N-terminus is not required for nucleation, it does affect the process when present and most mutations that accelerate nucleation are located in the N-terminus. Interestingly, the effects of mutations in residues immediately before position 17 suggest that the formation of a structural element in this region may interfere with nucleation.

That accelerated nucleation is a common cause of fAD is also supported by the effects of mutations in *APP* outside of Aß and by the effects of mutations in *PSEN1* and *PSEN2*. These mutations destabilise enzyme-substrate complexes, increasing the production of the longer Aß peptides that more effectively nucleates amyloid formation (*Szaruga et al., 2017*; *Veugelen et al., 2016*). In addition, Aß42 oligomers are hypothesised to be more toxic (*Michaels et al., 2020*; *Bolognesi et al., 2010*). It is possible that the effects of some of the mutations reported here on nucleation are also mediated by a change in the concentration of Aß rather than by an increase in a kinetic rate parameter. Some of the variants evaluated here may have additional effects, for example, altering cleavage of APP. Future work will be needed to test these hypotheses.

Comparing our results to the effects of mutations on Aß solubility quantified in a previous high-throughput analysis (*Gray et al., 2019*) provides evidence that, in the same type of cell (yeast), Aß can aggregate in at least two different ways. Moreover, the different performance of the two sets of scores from these datasets in classifying fAD mutations suggests that one of these aggregation processes (quantified by the nucleation assay employed here) is likely to be very similar to the aggregation that occurs in the human brain in fAD. The other pathway of aggregation (quantified by the solubility assay; *Gray et al., 2019*), however, is less obviously related to the human disease, because mutations that cause fAD do not consistently affect it. This second aggregation pathway is, at least to a large extent, driven by changes in hydrophobicity, similar to what we previously reported for the aggregation in yeast of the ALS protein, TDP-43 (*Bolognesi et al., 2019*).

More generally, our results highlight how the combination of deep mutational scanning and human genetics can be a general 'genetic' strategy to quantify the disease relevance of biological assays. Many in vitro and in vivo assays are proposed as 'disease models' in biomedical research with their relevance often justified by how 'physiological' the assays seem or how well phenotypes observed in the model match those observed in the human disease. The range of phenotypes that can be assessed and their similarity to the pathology of AD human brains are appealing features of many animal models of AD and many important insights have been derived – and will continue to be derived – from animal models (*Sasaguri et al., 2017*). However, there are applications where animal models cannot be realistically used, for example, for high-throughput compound screening for drug discovery and for testing hundreds or thousands of genetic variants of unknown significance. For these applications, in vitro or cell-based (*Pimenova and Goate, 2020*; *Veugelen et al., 2016*) assays are required and an important challenge is to evaluate the 'disease relevance' of different assays. Our study highlights an approach to achieve this, which is to use the complete set of known disease-

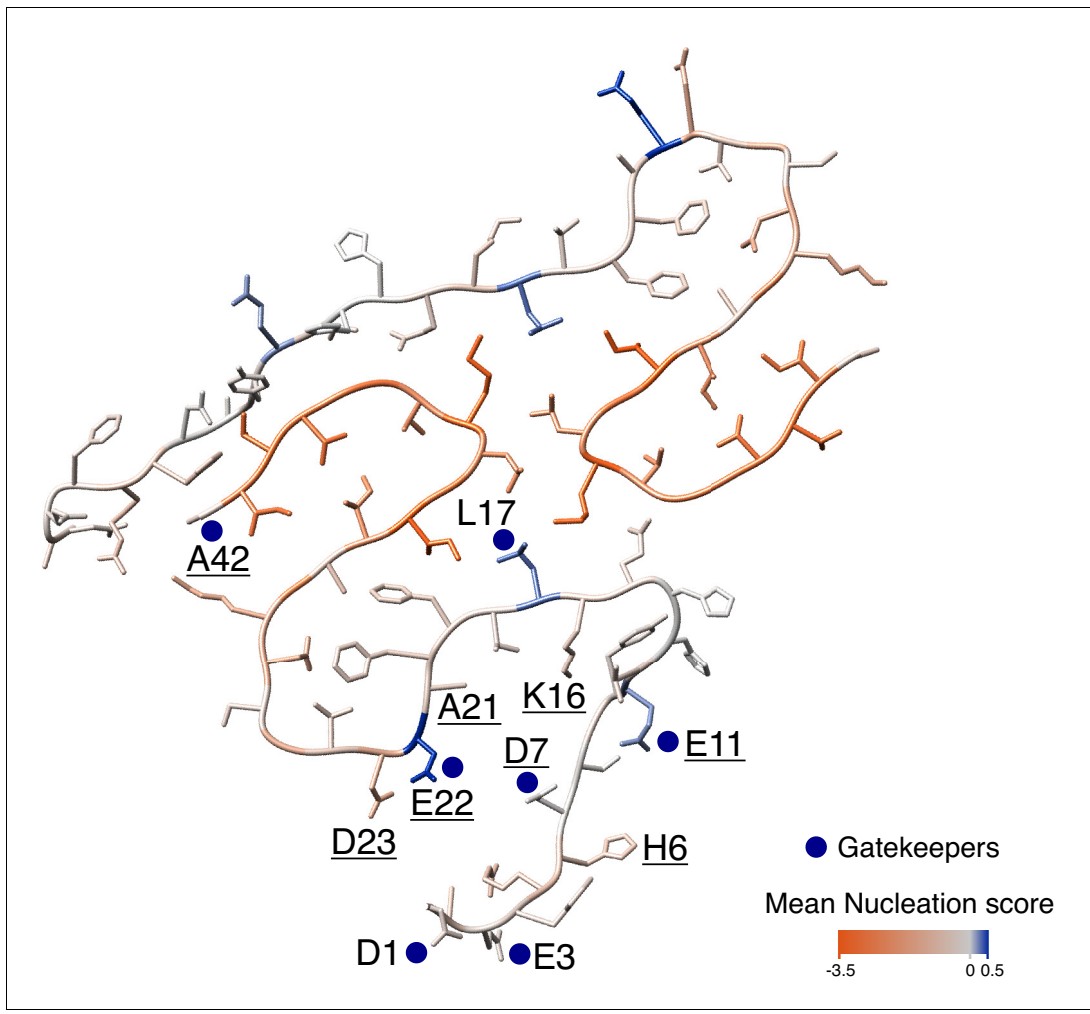

**Figure 5.** Mutational landscape of the amyloid beta (Aß) amyloid fibril. Average effect of mutations visualized on the cross-section of an Aß amyloid fibril (PDB accession 5KK3; *Colvin et al., 2016*). Nucleation gatekeeper residues and known familial Alzheimer's disease (fAD) mutations positions are indicated by the wild-type (WT) aa identity on one of the two monomers; gatekeepers are indicated with blue dots and fAD are underlined. A single layer of the fibril is shown and the unstructured N-termini (aa 1-14) are shown with different random coil conformations for the two Aß monomers. See *Figure 5—figure supplement 2* for alternative Aß42 amyloid polymorphs.

The online version of this article includes the following figure supplement(s) for figure 5:

**Figure supplement 1.** Modular organization of Aß42 and Aß40 polymorphs.

**Figure supplement 2.** Modular organization of mutational effects and gatekeepers visualized on Aß42 polymorphs.

causing mutations to quantify the 'genetic agreement' between an assay and a disease. Thus, although the yeast-based assay that we employed here might typically be dismissed as 'non-physiological,' 'artificial,' or 'lacking many features important for a neurological disease,' unbiased massively parallel genetic analysis provides very strong evidence that it is reporting on biopysicall events that are extremely similar to – or the same as – those that cause the human disease. Indeed, one could argue that this simple system is now better validated as a model of fAD than many others, including animal models where the effects of only one or a few mutations (including control mutations) have ever been tested. Similarly strong agreement between mutational effects in a cellular assay and the set of mutations already known to cause a disease is observed for other diseases (*Starita et al., 2017*; *Gelman et al., 2019*), suggesting the generality of this approach.

We suggest therefore that the combination of deep mutational scanning and human genetics provides a general strategy to quantify the disease relevance of in vitro and cell-based assays. We encourage that deep mutagenesis should be employed early in discovery programmes to 'genetically validate' (or invalidate) the relevance of assays for particular diseases. The concordance between mutational effects in an assay and a disease is an unbiased metric that can be used to prioritize between different assays. Quantifying the 'genetic agreement' between an assay and a disease will help prevent time and resources being wasted on research that actually has little relevance to a disease.

Finally, the strikingly consistent effects of the dominant fAD mutations in our assay further strengthen the evidence that fAD is a 'nucleation disease' ultimately caused by an increased rate of amyloid nucleation (*Aprile et al., 2017*; *Cohen et al., 2018*; *Knowles et al., 2009*). This accelerated nucleation can be caused by the direct effects of mutations in Aß — such as those quantified here — or by changes in upstream factors (*Szaruga et al., 2017*). If this hypothesis is correct, then nucleation is the key bioph step to target to prevent or treat AD. We suggest that the 'genetic validation' of assays by mutational scanning and comparison to sets of known disease-causing mutations will be increasingly important in assay development and drug discovery pipelines.

## Materials and methods

### Plasmid library construction

The plasmid $P_{CUP1}$-Sup35N-Aβ42 used in this study was a kind gift from the Chernoff lab (*Chandramowlishwaran et al., 2018*).

The Aβ coding sequence and two flanking regions of 52 bp and 72 bp, respectively, upstream and downstream of Aβ were amplified (primers MS_01 and MS_02, *Supplementary file 3*) by error-prone PCR (Mutazyme II DNA polymerase, Agilent). Thirty cycles of amplification and 0.01 ng of initial template were used to obtain a mutagenesis rate of 16 mutations/kb, according to the manufacturer's protocol. The product was treated with DpnI (FastDigest, Thermo Scientific) for 2 hr and purified by column purification (MinElute PCR Purification Kit, Qiagen). The fragment was digested with EcoRI and XbaI restriction enzymes (FastDigest, Thermo Scientific) for 1 hr at 37°C and purified from a 2% agarose gel (QIAquick Gel Extraction Kit, Qiagen). In parallel, the $P_{CUP1}$-Sup35N-Aβ42 plasmid was digested with the same restriction enzymes to remove the WT Aβ sequence, treated with alkaline phosphatase (FastAP, Thermo Scientific) for 1 hr at 37°C to dephosphorylate the 5' ends, and purified from a 1% agarose gel (QIAquick Gel Extraction Kit, Qiagen).

Mutagenised Aβ was then ligated into the linearised plasmid in a 5:1 ratio (insert:vector) using a ligase treatment (T4, Thermo Scientific) overnight. The reaction was dialysed with a membrane filter (Merck Millipore) for 1 hr, concentrated 4x, and transformed in electrocompetent *Escherichia coli* cells (10-beta Electrocompetent, NEB). Cells were recovered in SOC medium and plated on LB with ampicillin. A total of 4.1 million transformants were estimated, ensuring that each variant of the library was represented more than 10 times; 50 ml of overnight *E. coli* culture was harvested to purify the Aβ plasmid library with a midi prep (Plasmid Midi Kit, Qiagen). The resulting library contained 29.9% of WT Aβ, 23.8% of sequences with 1 nt change, and 21.8% of sequences with 2 nt changes.

### Large-scale yeast transformation

*Saccharomyces cerevisiae* [psi-pin-] (*MATa ade1-14 his3 leu2-3,112 lys2 trp1 ura3-52*) strain (also provided by the Chernoff lab) was used in all experiments in this study (*Chandramowlishwaran et al., 2018*).

Yeast cells were transformed with the Aβ plasmid library starting from an individual colony for each transformation tube. After an overnight pre-growth culture in YPDA medium at 30°C, cells were diluted to $OD_{600}$ = 0.3 in 175 ml YPDA and incubated at 30°C 200 rpm for ~5 hr. When cells reached the exponential phase, they were harvested, washed with milliQ, and resuspended in sorbitol mixture (100 mM LiOAc, 10 mM Tris pH 8, 1 mM EDTA, 1M sorbitol). After a 30 min incubation at room temperature (RT), 5 µg of plasmid library and 175 µl of ssDNA (UltraPure, Thermo Scientific) were added to the cells. PEG mixture (100 mM LiOAc, 10 mM Tris pH 8, 1 mM EDTA pH 8, 40% PEG3350) was also added and cells were incubated for 30 min at RT and heat-shocked for 15 min at

42°C in a water bath. Cells were harvested, washed, resuspended in 350 ml recovery medium (YPD, sorbitol 0.5M, 70 mg/L adenine) and incubated for 1.5 hr at 30°C 200 rpm. After recovery, cells were resuspended in 350 ml -URA plasmid selection medium and allowed to grow for 50 hr. Transformation efficiency was calculated for each tube of transformation by plating an aliquot of cells in -URA plates. Between 1 and 2.5 million transformants per tube were obtained. Two days after transformation, the culture was diluted to $OD_{600}$ = 0.02 in 1 l -URA medium and grown until the exponential phase. At this stage, cells were harvested and stored at −80°C in 25% glycerol.

## Selection experiments

Three independent replicate selection experiments were performed. Tubes were thawed from the −80°C glycerol stocks and mixed proportionally to the number of transformants in a 1 l total -URA medium at $OD_{600}$ = 0.05. A minimum of 3.7 million yeast transformants were used for each replicate to ensure the coverage of the full library and reaching therefore a 10x coverage of each variant.

Once the culture reached the exponential phase, cells were resuspended in 1 l protein inducing medium (-URA, 20% glucose, 100 µM $Cu_2SO_4$) at $OD_{600}$ = 0.05. As a result, each variant was represented at least 100 times at this stage. After 24 hr the input pellets were collected by centrifuging 220 ml of cells and stored at −20°C for later DNA extraction (input pellets). In parallel, 18.5 million cells of the same culture underwent selection, with a starting coverage of at least 50 copies of each variant in the library. For selection, cells were plated on -ADE-URA selective medium in 145 cm$^2$ plates (Nunc, Thermo Scientific) and let grow for 7 days at 30°C. Colonies were then scraped off the plates and recovered with PBS 1x to be centrifuged and stored at −20°C for later DNA extraction (output pellets).

For individual testing of specific variants, cells were plated on -URA (control) and -ADE-URA (selection) plates in three independent replicates. Individual growth was calculated as the percentage of colonies growing -ADE-URA relative to colonies growing in -URA.

## DNA extraction

The input and output pellets (three replicates, six tubes in total) were thawed and resuspended in 2 ml extraction buffer (2% Triton-X, 1% SDS, 100 mM NaCl, 10 mM Tris pH 8, 1 mM EDTA pH 8), and underwent two cycles of freezing and thawing in an ethanol-dry ice bath (10 min) and at 62°C (10 min). Samples were then vortexed together with 1.5 ml of phenol:chloroform:isoamyl 25:24:1 and 1.5 g of glass beads (Sigma). The aqueous phase was recovered by centrifugation and mixed again with 1.5 ml phenol:chloroform:isoamyl 25:24:1. DNA precipitation was performed by adding 1:10 V of 3M NaOAc and 2.2 V of 100% cold ethanol to the aqueous phase and incubating the samples at −20°C for 1 hr. After a centrifugation step, pellets were dried overnight at RT.

Pellets were resuspended in 1 ml resuspension buffer (10 mM Tris pH 8, 1 mM EDTA pH 8) and treated with 7.5 µl RNase A (Thermo Scientific) for 30 min at 37°C. The DNA was finally purified using 75 µl of silica beads (QIAEX II Gel Extraction Kit, Qiagen), washed and eluted in 375 µl elution buffer.

DNA concentration in each sample was measured by quantitative PCR, using primers (MS_03 and MS_04, *Supplementary file 3*) that anneal to the origin of replication site of the plasmid at 58°C.

## Sequencing library preparation

The library was prepared for high-throughput sequencing in two rounds of PCR (Q5 High-Fidelity DNA Polymerase, NEB). In PCR1, the Aβ region was amplified for 15 cycles at 68°C with frame-shifted primers (MS_05 to MS_18, *Supplementary file 3*) with homology to Illumina sequencing primers; 300 million of molecules were used for each input or output sample. The products of PCR1 were purified with an ExoSAP-IT treatment (Affymetrix) and a column purification step (QIAquick PCR Purification Kit) and then used as the template of PCR2. This PCR was run for 10 cycles at 62°C with Illumina indexed primers (MS_19 to MS_25, *Supplementary file 2*) specific for each sample (three inputs and three outputs). The six samples were then pooled together equimolarly. The final library sample was purified from a 2% agarose gel with silica beads (QIAEX II Gel Extraction Kit, Qiagen); 125 bp paired-end sequencing was run on an Illumina HiSeq2500 sequencer at the CRG Genomics Core Facility.

## Data processing

FastQ files from paired-end sequencing of the Aß library before ('input') and after selection ('output') were processed using a custom pipeline (https://github.com/lehner-lab/DiMSum). DiMSum (*Faure et al., 2020*) is an R package that uses different sequencing processing tools such as FastQC (http://www.bioinformatics.babraham.ac.uk/projects/fastqc/) (for quality assessment), Cutadapt (*Martin, 2011*) (for constant region trimming), and USEARCH (*Edgar, 2010*) (for paired-end read alignment). Sequences were trimmed at 5′ and 3′, allowing an error rate of 0.2 (i.e., read pairs were discarded if the constant regions contained more than 20% mismatches relative to the reference sequence). Sequences differing in length from the expected 126 bp or with a Phred base quality score below 30 were discarded. As a result of this processing, around 150 million total reads passed the filtering criteria.

At this stage, unique variants were aggregated and counted using Starcode (https://github.com/gui11aume/starcode). Variants containing indels and nonsynonymous variants with synonymous substitutions in other codons were excluded. The result is a table of variant counts which can be used for further analysis.

For downstream analysis, variants with less than 50 input reads in any of the replicates were excluded and only variants with a maximum of two aa mutations were used.

## Nucleation scores and error estimates

On the basis of variant counts, the DiMSum pipeline (*Faure et al., 2020*; https://github.com/lehner-lab/DiMSum) was used to calculate nucleation scores (NS) and their error estimates. For each variant in each replicate NS was calculated as:

$$Nucleation\,score = ES_i - ES_{wt}$$

where $ES_i = log(F_i\,OUTPUT) - log(F_i\,INPUT)$ for a specific variant and $ES_{wt} = log(F_{wt}\,OUTPUT) - log(F_{wt}\,INPUT)$ for Aß WT.

DiMSum models measurement error of NS by assuming that variants with similar counts in input and output samples have similar errors. Based on errors expected from Poisson-distributed count data, replicate-specific additive and multiplicative (one each for input and output samples) modifier terms are fit to best describe the observed variance of NS across all variants simultaneously.

After error calculation, NS were merged by using the error-weighted mean of each variant across replicates and centered using the error-weighted means frequency of synonymous substitutions arising from single nt changes. Merged NS and NS for each independent replicate, as well as their associated error estimates, are available in *Supplementary file 4*.

Nonsense (stop) mutants were excluded for the analysis except when indicated (*Figure 2A and C* and *Figure 2—figure supplement 1A*).

## K-medoids clustering

We used K-medoids, or the partitioning around medoids algorithm, to cluster the matrix of single aa variant NS estimates by residue position with the number of clusters estimated by optimum average silhouette width, for values of K in [1,10]. The silhouette width is a measure of how similar each object (in this case residue position) is to its own cluster. In order to take into account uncertainty in NS estimates in the determination of the optimum number of clusters, we repeated this analysis after random resampling from the NS (error) distributions of each single aa variant (n = 100). Based on this clustering, we defined the N-terminus as aa 2-26 and the C-terminus as aa 27-41 (*Figure 2—figure supplement 1B*). Seven positions where as many (or more) single mutations increase as decrease nucleation were defined as 'gatekeepers' (D1, E3, D7, E11, L17, E22, A42) and excluded from the N- and C-terminus classes. Only those positions where most mutations are significantly different from WT (FDR = 0.1) were considered for the definition of gatekeepers.

## Aa properties, aggregation, and variant effect predictors

Nucleation scores were correlated with aa properties and scores from aggregation, solubility, and variant effect prediction algorithms. Pearson correlations were weighted based on the error terms associated with the NS of each variant using the R package 'weights.' The aa property features were retrieved from a curated collection of numerical indices representing various aa physicochemical and

biochemical properties (http://www.genome.jp/aaindex/). We also used a principal component of these aa properties from a previous work (PC1; *Bolognesi et al., 2019*) that relates strongly to changes in hydrophobicity. For each variant (single and double aa mutants), the values of a specific aa property represent the difference between the mutant and the WT scores.

For the aggregation and solubility algorithms (Tango [*Fernandez-Escamilla et al., 2004*], Zyggregator [*Tartaglia and Vendruscolo, 2008*], CamSol [*Sormanni et al., 2015*], and Waltz [*Oliveberg, 2010*]), individual residue-level scores were summed to obtain a score per aa sequence. We then calculated the log value for each variant relative to the WT score (single and double aa mutants for Tango, Zyggregator, CamSol and single aa mutants for Waltz). For the variant effect predictors (Polyphen [*Adzhubei et al., 2013*] and CADD [*Rentzsch et al., 2019*]), we also calculated the log value for each variant (only single aa mutants) but in this case values were scaled relative to the lowest predicted score.

### fAD, gnomAD, and Clinvar variants

The table of fAD mutations used in this study was taken from https://www.alzforum.org/mutations/app. Allele frequencies of *APP* variants were retrieved from gnomAD (*Karczewski, 2020*) (https://gnomad.broadinstitute.org/) and the clinical significance of variants was taken from their Clinvar (*Landrum et al., 2014*) classification (https://www.ncbi.nlm.nih.gov/clinvar).

ROC curves were built and AUC values were obtained using the 'pROC' R package.

### PDB structures

The coordinates of the following PDB structures were used for *Figure 5*, *Figure 5—figure supplements 1* and *2*: 5OQV, 2NAO, 5KK3, 2BEG, 2MXU, 5AEF, 6SHS, 2LMN, 2LMP, 2LNQ, 2MVX, 2M4J, 2MPZ (*Gremer et al., 2017*; *Colvin et al., 2016*; *Wälti et al., 2016*; *Lührs et al., 2005*; *Xiao et al., 2015*; *Schmidt et al., 2015*; *Kollmer et al., 2019*; *Lu et al., 2013*; *Qiang et al., 2012*; *Sgourakis et al., 2015*; *Schütz et al., 2015*).

## Acknowledgements

Work in the lab of BB is supported by the Spanish Ministry of Science, Innovation and Universities through the project RTI2018-101491-A-I00 (MICIU/FEDER), by the CERCA Program/Generalitat de Catalunya and by funding from the Agencia de Gestio d'Ajuts Universitaris i de Recerca (2019FI_B 01311) to MS Work in the lab of BL is supported by a European Research Council (ERC) Consolidator grant (616434), the Spanish Ministry of Science, Innovation and Universities (BFU2017-89488-P and SEV-2012–0208), the Bettencourt Schueller Foundation, Agencia de Gestio d'Ajuts Universitaris i de Recerca (AGAUR, 2017 SGR 1322.), and the CERCA Program/Generalitat de Catalunya. We acknowledge the support of the Spanish Ministry of Science and Innovation to the EMBL partnership and the Centro de Excelencia Severo Ochoa. We thank the Chernoff lab for kindly providing strains and plasmids and the CRG Genomics core facility for their assistance with sequencing.

## Additional information

### Funding

| Funder | Grant reference number | Author |
|---|---|---|
| Ministerio de Ciencia e Innovación | RTI2018-101491-A-I00 | Benedetta Bolognesi |
| Ministerio de Ciencia e Innovación | BFU2017-89488-P | Ben Lehner |
| H2020 European Research Council | 616434 | Ben Lehner |
| Agència de Gestió d'Ajuts Universitaris i de Recerca | SGR 1322 | Ben Lehner |
| Agència de Gestió d'Ajuts Universitaris i de Recerca | 2019FI_B 01311 | Mireia Seuma |

Fondation Bettencourt Schuel-    Prize                      Ben Lehner
ler

The funders had no role in study design, data collection and interpretation, or the
decision to submit the work for publication.

## Author contributions

Mireia Seuma, Conceptualization, Formal analysis, Validation, Investigation, Visualization, Methodology, Writing - original draft, Writing - review and editing; Andre J Faure, Software, Investigation, Visualization, Methodology; Marta Badia, Investigation; Ben Lehner, Conceptualization, Supervision, Funding acquisition, Writing - original draft, Writing - review and editing; Benedetta Bolognesi, Conceptualization, Supervision, Methodology, Funding acquisition, Writing - original draft, Writing - review and editing

## Author ORCIDs

Mireia Seuma (iD) https://orcid.org/0000-0002-6140-4530
Andre J Faure (iD) https://orcid.org/0000-0002-4471-5994
Marta Badia (iD) https://orcid.org/0000-0002-9712-9163
Ben Lehner (iD) https://orcid.org/0000-0002-8817-1124
Benedetta Bolognesi (iD) https://orcid.org/0000-0002-6632-947X

## Decision letter and Author response

Decision letter https://doi.org/10.7554/eLife.63364.sa1
Author response https://doi.org/10.7554/eLife.63364.sa2

# Additional files

## Supplementary files

• Supplementary file 1. Table listing the impact on aggregation rates for 16 Aß42 variants for which these measurements could be retrieved from the literature. For the same variants, the table also reports nucleation scores, as quantified in this study, and the qualitative agreement or disagreement with the previously published data.

• Supplementary file 2. Table listing the mutations in Aß42 that significantly increase nucleation score and that are therefore proposed as novel familial Alzheimer's disease (fAD) candidates. For each mutation, the corresponding nucleation score (NS) is reported.

• Supplementary file 3. List of oligonucleotides used in this study.

• Supplementary file 4. Processed data required to make all analyses and figures in this paper. Read counts, nucleation scores, and associated error terms are reported for each Aß42 variant in each replicate. See sheet one for a deeper explanation of headers.

• Transparent reporting form

## Data availability

Raw sequencing data and the processed data table (Supplementary file 4) have been deposited in NCBI's Gene Expression Omnibus (GEO) as record GSE151147. All code used for data analysis is available at https://github.com/BEBlab/abeta (copy archived at https://archive.softwareheritage.org/swh:1:rev:86e1e1be4ee6eb97c1c00b0bd53f98f4e4ea807f/).

The following dataset was generated:

| Author(s) | Year | Dataset title | Dataset URL | Database and Identifier |
|---|---|---|---|---|
| Seuma M, Faure A, Badia M, Lehner B, Bolognesi B | 2020 | The genetic landscape for amyloid beta fibril nucleation accurately discriminates familial Alzheimer's disease mutations | https://www.ncbi.nlm.nih.gov/geo/query/acc.cgi?acc=GSE151147 | NCBI Gene Expression Omnibus, GSE151147 |

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
