## [Decision Letter]

**Acceptance summary:**

This paper describes an in vitro nucleation assay based on the fusion of Aβ mutants with yeast sup35. They screened almost 14500 mutants and found only 16% to increase nucleation, while others suppress the process. Importantly, they define modular regions of Aβ linked to various aggregation properties and the assay also finds known familial AD mutations. Hence, this work reports a rapid means to assign pathogenicity to Aβ variants in high throughput screens and will help in assigning function to variants of unknown significance.

**Decision letter after peer review:**

Thank you for submitting your article "The genetic landscape for amyloid beta fibril nucleation accurately discriminates familial Alzheimer's disease mutations" for consideration by *eLife*. Your article has been reviewed by three peer reviewers, and the evaluation has been overseen by a Reviewing Editor and Huda Zoghbi as the Senior Editor. The reviewers have opted to remain anonymous.

Summary:

This paper reports a deep scanning mutagenesis screening to test the effects of over 14,000 single and double mutations in Aβ42, the proteolytic product from the amyloid precursor protein associated with Alzheimer disease, on the aggregation properties of the peptide. In contrast to previous work, the current study is based on an aggregation gain-of-function screen, where a construct expressing Aβ42 fused to yeast sup35 is used to trigger prion conversion of full length sup35. The authors make a number of sensible arguments (but also some overstatements) that differentiate their assay from that used in previous work. The findings include the identification of mutations that enhance or suppress Aβ nucleation as well as modular regions in the peptide. The assay also is able to identify the fAD mutations and the conclusion is therefore that this would be a great assay for quick assessment of pathogenicity to Aβ variants of unknown significance. The reviewers are enthusiastic about the approach, but also pointed out several issues that need to be addressed.

Essential Revisions:

1) Overstatements related to disease relevance and in particular the related comments listed below from reviewers 1 and 3:

a) The manuscript states that it is not clear why APP/Aβ mutations cause AD, but it would be good to see a discussion of recent mechanistic work demonstrating that fAD mutations destabilise APP-GSEC complexes, resulting in the release of longer Aβ peptides (PMID: 28753424). Thus, the mutations could be shifting the profile of Aβ species produced as well as influencing aggregation, this should be discussed.

b) The conclusions that "the rate of nucleation in a cell-based assay accurately identifies mutations that cause dominant familial Alzheimer's disease" and "AD is actually a nucleation disease" should be reviewed, while considering that several other “very aggressive” mutations in APP and in Presenilins (γ secretase) do not generated mutant, but wild type Aβ peptides. In fact, the most aggressive mutations in APP (such as the Iberian or Autrian) and mutations in Presenilin share a common mechanism that results in the enhanced generation of longer Aβ peptides (Szaruga et al., 2017). How do the authors incorporate these findings into their model? In addition, it is known that pathogenic mutations in Aβ1-42 do not only affect its aggregation propensity but also the cleavage of APP by β-secretase. The protective, Icelandic APP variant is one of these cases.

In short, the Discussion would benefit from a broader perspective.

2) Potential bias of the assay itself: coupling Aβ aggregation to sup35 aggregation and immobilizing the N-terminal part of Aβ in a linker (from reviewer 2):

a) The specific purpose of this work is to map the genetic landscape for nucleation of Aβ42. Yet the reporter used here is the ability of a fragment of another amyloid (prion) protein sup35 to nucleate endogenous sup35. This is a totally different aggregation prone sequence than Aβ (Q/N rich instead of hydrophobic) with different intrinsic kinetics. What are the controls guaranteeing that a)Aβ42 is indeed rate limiting, b) that sup35 seeding is scaling along with Aβ42 nucleation (some Aβ mutations might lead to many small seeds while other to less but bigger seeds) and c) that both Aβ42 and sup35N domain aggregation are independent of each other in this fusion. How can antagonistic or synergistic effects be excluded?

b) The sup35 fragment is N-terminally fused to Aβ42 and the data show biggest effects on the aggregation propensity at the C-terminus. Is the fact that the 16-23 fragment is less prominent not a bias resulting of transforming the N-terminal unstructured part of Aβ42 into a linker in the fusion thereby underestimating effect towards the N terminal part of the domain?

c) The authors validate their assay by showing that previously measured nucleation rates of disease mutants correlate with their “nucleation” enrichment scores. The problem is that this only represents a handful of mutants (5 mutants) the majority of which are situated along positions 21/22/23 of the Aβ peptide. There is no guarantee that mutants in very different contexts such as in the flexible N-terminal part of the region or the C-terminal aggregation prone region will respond in such a nicely correlated manner.

3) Overstatements in the Discussion that we hope can be addressed by making textual adaptations (all reviewers):

a) The authors conclude that their results are more relevant to disease because they better report on nucleation than the study of Gray et al. They also conclude that they report on another mode of aggregation. I don't think there is evidence for that in this manuscript. First, although nucleation plays an important role in amyloid diseases this is not necessarily all of it. Second, the study of Gray et al. -although not correlating well with nucleation data- still identifies the importance of the 16-23 region and the C-terminal region in a more balanced manner. Both region are known to be crucial in determining Aβ nucleation.

b) I find the Discussion to be unnecessarily dismissive of animal models and other model systems. Particularly "this simple system is now better-validated as a model of fAD than any other, including animal models where the effects of only one or a few mutations (including control mutations) have ever been tested.". The current paper models only one aspect of Aβ biology, aggregation, and doesn't take into account Aβ generation nor the mechanisms linking Aβ to neurodegeneration. Thus I think this required rewriting to acknowledge that each model has its strengths/limitations

c) Likewise there is an omission of other cell based and cell free models that have been used for assigning significance to pathological variants and should be incorporated into the Discussion, e.g. PMID: 32032730, PMID: 27100199

d) The statement "Many in vitro and in vivo assays are proposed as “disease models” in biomedical research with their relevance often justified by how “physiological” the assays seem or how well phenotypes observed in the model match those observed in the human disease. However, such criteria are largely subjective, and assays that seem relevant to a disease may actually turn out to be reporting on irrelevant biochemical events, resulting in the of drugs that then fail in clinical trials." is vague – can the authors give specific examples and references with relevance to AD?

We believe that the other comments are straightforward to address with further changes to the text.

Reviewer #1:

In the presented manuscript Seuma et al. describe the analysis of the effects of >14000 single or double mutations in Aβ42 on the aggregation properties of the peptide. Using a yeast cell-based assay, the authors determined nucleation scores for the tested Aβ42 mutant peptides and defined several positions and domains that play distinct roles in the aggregation process.

The mutational analysis reveals a strong bias towards reduced Aβ nucleation, with only 16% of the mutants increasing nucleation. It is interesting to see that the data seem to differentiate solubility/hydrophobicity and nucleation, suggesting that two factors may contribute to Aβ aggregation.

The data also indicate that mutations that decrease nucleation are enriched in the C-terminus of Aβ while mutations that increase it are enriched in the N-terminus, suggesting that the formation of a hydrophobic C-terminal core mediates nucleation. Furthermore, the data reveal that decreasing both the net charge of the peptide and the total number of charged residues increases nucleation. In fact, most of the negative charges present in Aβ42 are proposed to be “gatekeepers” of nucleation.

Notably, mutating the negatively charged gatekeepers to the polar, positively charged or small side chain (G and A) amino acids increases nucleation; while substituting them for hydrophobic residues generally reduced nucleation. The authors thus proposed a model in which "the negative charge at these positions acts to limit nucleation, but that the overall polar and unstructured nature of the N-terminus must be maintained for effective nucleation.".

Finally, the authors concluded that the Aβ nucleation accurately discriminates familial AD (fAD) mutations. However, a better presentation and case-by-case analysis of the fAD data is required to fully assess the disease relevance of the observations.

• The selection assay (reporting on nucleation) seems to be highly reproducible, as supported by data presented in Figure 1B. Nevertheless, this reviewer wonders about the potential impact of the Sup35N fusion on Aβ solubility/hydrophobicity and aggregation. In addition, Aβ aggregation has been reported to be concentration dependent and it is not clear from the data whether the mutant peptides are, or not, expressed at similar levels.

Could peptide fusion and/or expression levels have changed the intrinsic attributes of Aβ42 and could these potential alterations explain -at least to some degree- the mismatch observed between nucleation and hydrophobicity (Figure 1D, 3F)?

• Figure 2 presents the effects of the amino acid changes on the nucleation of Aβ. The authors mention that all possible amino acid changes in Aβ42 were introduced; however, there are several white spots corresponding to non-existing mutants. Why these mutants are missing?

• The authors proposed and discussed the presence of gatekeepers of nucleation. How precisely these have been selected? Is there any cut-off score? Data in Figure 2D does not clarify this point: data for H13 or Q15 do not seem to be much different from the positions selected. Furthermore, only few mutations seem to have been tested at position A42 and an equal number of mutations increase or decrease nucleation, is this residue/position truly a gatekeeper?

• In the last part of the manuscript, the authors analysed the nucleation of fAD-linked mutations in Aβ. The figures display "ROC curves built using 12 fAD mutants versus all other single aa mutants in the 484 datasets for variant effect and aggregation predictors". This reviewer would appreciate a figure where data for each specific fAD variant is displayed; or alternatively a table presenting the key data from a case by case analysis. It would be also interesting to see how the data relates to clinical onset.

• The conclusions that "the rate of nucleation in a cell-based assay accurately identifies mutations that cause dominant familial Alzheimer's disease" and "AD is actually a nucleation disease" should be reviewed, while considering that several other “very aggressive” mutations in APP and in Presenilins (γ secretase) do not generated mutant, but wild type Aβ peptides. In fact, the most aggressive mutations in APP (such as the Iberian or Autrian) and mutations in Presenilin share a common mechanism that results in the enhanced generation of longer Aβ peptides (Szaruga et al., 2017). How do the authors incorporate these findings into their model?

In addition, it is known that pathogenic mutations in Aβ1-42 do not only affect its aggregation propensity but also the cleavage of APP by β-secretase. The protective, Icelandic APP variant is one of these cases.

In short, the Discussion would benefit from a broader perspective.

• The authors should note that the pathogeneicity of the H6R is unclear.

Reviewer #2:

This study reports a deep scanning mutagenesis screening mapping the genetic landscape of Aβ42 fibril nucleation.

Another deep scanning mutagenesis study on Aβ42 was recently published (Gray et al., 2019).

While Gray et al. use a loss of function assay of an Aβ42-DHFR fusion, Seuma et al. here use an aggregation gain-of-function screen whereby an Aβ42 fusion to the N-domain of sup35 is used to trigger prion conversion of endogenous full length sup35.

The main claim of the current manuscript is that the sup35 reporter assay allows to better map sequence determinants of Aβ42 nucleation while the study by Gray et al. reports more on Aβ42 solubility. The authors further claim that their approach reports on another mode of aggregation that is more relevant to disease. This based on the fact that the current data correlate with the nucleation rates of previously reported Aβ disease mutants (12 mutants) while the results of Gray et al. do not correlate with nucleation rates but with hydrophobicity.

While the actual data analysis is very interesting and often plausible I still have major concerns about the potential for biases in the current experimental setup and therefore of the reliability of the findings along the entire sequence. Clarifying these issues will require essential experimental validation.

More specifically:

1) The specific purpose of this work is to map the genetic landscape for nucleation of Aβ42. Yet the reporter used here is the ability of a fragment of another amyloid (prion) protein sup35 to nucleate endogenous sup35. This is a totally different aggregation prone sequence than Aβ (Q/N rich instead of hydrophobic) with different intrinsic kinetics. What are the controls guaranteeing that a)Aβ42 is indeed rate limiting, b) that sup35 seeding is scaling along with Aβ42 nucleation (some Aβ mutations might lead to many small seeds while other to less but bigger seeds) and c) that both Aβ42 and sup35N domain aggregation are independent of each other in this fusion. How can antagonistic or synergistic effects be excluded?

2) The sup35 fragment is N-terminally fused to Aβ42 and the data show biggest effects on the aggregation propensity at the C-terminus. Is the fact that the 16-23 fragment is less prominent not a bias resulting of transforming the N-terminal unstructured part of Aβ42 into a linker in the fusion thereby underestimating effect towards the N terminal part of the domain?

3) The authors validate their assay by showing that previously measured nucleation rates of disease mutants correlate with their “nucleation” enrichment scores. The problem is that this only represents a handful of mutants (5 mutants) the majority of which are situated along positions 21/22/23 of the Aβ peptide. There is no guarantee that mutants in very different contexts such as in the flexible N-terminal part of the region or the C-terminal aggregation prone region will respond in such a nicely correlated manner.

Overall therefore the absence of bias in the current experimental setup needs to be addressed by a more rigorous validation. Mutants that increase/decrease nucleation along different parts of the sequence should be experimentally validated in the same manner than the Yang et al., 2018 paper currently used to show correlation.

Finally, the authors conclude that their results are more relevant to disease because they better report on nucleation than the study of Gray et al. They also conclude that they report on another mode of aggregation. I don't think there is evidence for that in this manuscript. First, although nucleation plays an important role in amyloid diseases this is not necessarily all of it. Second, the study of Gray et al. -although not correlating well with nucleation data- still identifies the importance of the 16-23 region and the C-terminal region in a more balanced manner. Both region are known to be crucial in determining Aβ nucleation.

Reviewer #3:

In this manuscript, the authors use an in vitro nucleation assay to assess the aggregation properties of 14 483 mutated forms of Aβ, encompassing all possible single amino acid changes as well as double amino acid mutations. They identify mutations that both enhance and suppress Aβ nucleation, and define modular regions of Aβ linked to various aggregation properties. Importantly, the assay is able to identify known fAD mutations. Based on these results, the authors conclude that their system provides a rapid, cost effective means to assign pathogenicity to Aβ variants of unknown significance.

Although I find the work to be technically sound, and to provide interesting insights into the biochemical properties of Aβ, I think the conclusions, particular with regards to disease relevance, to be somewhat overstated in part and there are some omissions of relevant literature. Some specific points are listed below:

1) The authors state that "Moreover, given the human mutation rate and population size, it is likely that nearly all of these possible variants in Aβ actually exist in at least one individual currently alive on the planet" – however, these mutations may not be compatible with life, and as the majority (14015) examined are double mutations, how likely are these to exist in people? I am not sure that the species with double amino acid alterations bear relevance to disease or are likely to exist in individuals

2) The manuscript states that it is not clear why APP/Aβ mutations cause AD, but it would be good to see a discussion of recent mechanistic work demonstrating that fAD mutations destabilise APP-GSEC complexes, resulting in the release of longer Aβ peptides (PMID: 28753424). Thus, the mutations could be shifting the profile of Aβ species produced as well as influencing aggregation, this should be discussed.

3) Relating to point 2, I think the Introduction should include a description of the tripeptide cleavage pathways that result in multiple forms of Aβ, and also recent work that Aβ 43 is more neurotoxic and aggregation prone than 42.

4) I find the Discussion to be unnecessarily dismissive of animal models and other model systems. Particularly "this simple system is now better-validated as a model of fAD than any other, including animal models where the effects of only one or a few mutations (including control mutations) have ever been tested.". The current paper models only one aspect of Aβ biology, aggregation, and doesn't take into account Aβ generation nor the mechanisms linking Aβ to neurodegeneration. Thus I think this required rewriting to acknowledge that each model has its strengths/limitations

5) Likewise there is an omission of other cell based and cell free models that have been used for assigning significance to pathological variants and should be incorporated into the Discussion, e.g. PMID: 32032730, PMID: 27100199

6) The statement "Many in vitro and in vivo assays are proposed as “disease models” in biomedical research with their relevance often justified by how “physiological” the assays seem or how well phenotypes observed in the model match those observed in the human disease. However, such criteria are largely subjective, and assays that seem relevant to a disease may actually turn out to be reporting on irrelevant biochemical events, resulting in the of drugs that then fail in clinical trials." is vague – can the authors give specific examples and references with relevance to AD?

---

## [Author Response]

Essential Revisions:1) Overstatements related to disease relevance and in particular the related comments listed below from reviewers 1 and 3:a) The manuscript states that it is not clear why APP/Aβ mutations cause AD, but it would be good to see a discussion of recent mechanistic work demonstrating that fAD mutations destabilise APP-GSEC complexes, resulting in the release of longer Aβ peptides (PMID: 28753424). Thus, the mutations could be shifting the profile of Aβ species produced as well as influencing aggregation, this should be discussed.

We have added the following sentences to the text:

“Several mutations in *PSEN1* and *PSEN2*, the genes coding for the secretases performing sequential cleavage of APP, also lead to autosomal dominant forms of AD.”

“That accelerated nucleation is a common cause of fAD is also supported by the effects of mutations in *APP* outside of Aß42 and by the effects of mutations in *PSEN1* and *PSEN2*. These mutations destabilize enzyme-substrate complexes, increasing the production of the longer Aß42 peptide that more effectively nucleates amyloid formation (Szaruga et al., 2017; Veugelen et al., 2016). In addition, Aß42 oligomers are hypothesized to be more toxic (Michaels et al., 2020; Bolognesi et al., 2010). It is possible that the effects of some of the mutations reported here on nucleation are also mediated by a change in the concentration of Aß42 rather than by an increase in a kinetic rate parameter. In addition, some of the variants evaluated here may have additional effects, for example altering cleavage of APP. Future work will be needed to test these hypotheses.”

b) The conclusions that "the rate of nucleation in a cell-based assay accurately identifies mutations that cause dominant familial Alzheimer's disease" and "AD is actually a nucleation disease" should be reviewed, while considering that several other “very aggressive” mutations in APP and in Presenilins (γ secretase) do not generated mutant, but wild type Aβ peptides. In fact, the most aggressive mutations in APP (such as the Iberian or Autrian) and mutations in Presenilin share a common mechanism that results in the enhanced generation of longer Aβ peptides (Szaruga et al., 2017). How do the authors incorporate these findings into their model? In addition, it is known that pathogenic mutations in Aβ1-42 do not only affect its aggregation propensity but also the cleavage of APP by β-secretase. The protective, Icelandic APP variant is one of these cases.In short, the Discussion would benefit from a broader perspective.

We agree and have expanded the Introduction and the Discussion of our results.

The importance of accelerated nucleation as the cause of fAD is actually also in line with the proposed mechanism explaining the effect of fAD mutations beyond Aβ, such as the ones in *PSEN1*, *PSEN2*, and in *APP* outside of the Aβ region. A common effect of these mutations is to enhance the generation of Aβ42 over shorter versions of the peptide (Szaruga et al., 2017), therefore increasing Aβ42 relative and/or absolute concentration and facilitating nucleation. In addition, the substrate-enzyme destabilization caused by some mutations in PSEN1 and PSEN2 leads to an increase in the representation of longer Aβ peptides (≥Aβ42) displaying increased nucleation propensity and increased neurotoxicity (Veugelen et al., 2016; Benitova et al., 2012; Vandersteen et al., 2012; Conicella et al., 2014). We cannot exclude that also some of the mutations in our library, especially those in the very first or last residues of the peptide, could impact cleavage of APP in humans and lead to over-representation of longer Aβ peptides.

We have added the following sentences to the text:

“Several mutations in *PSEN1* and *PSEN2*, the genes coding for the secretases performing sequential cleavage of APP, also lead to autosomal dominant forms of AD.”

“The other variant, A2V, does not have a dominant effect on nucleation in our assay (Figure 2C), consistent with a recessive pattern of inheritance and a different mechanism of action, such as reduced ß-cleavage and increased Aß 42 generation, as previously proposed (Benilova et al., 2014).”

“That accelerated nucleation is a common cause of fAD is also supported by the effects of mutations in *APP* outside of Aß42 and by the effects of mutations in *PSEN1* and *PSEN2*. These mutations destabilize enzyme-substrate complexes, increasing the production of the longer Aß42 peptide that more effectively nucleates amyloid formation (Szaruga et al., 2017; Veugelen et al., 2016). In addition, Aß42 oligomers are hypothesized to be more toxic (Michaels et al., 2020; Bolognesi et al., 2010). It is possible that the effects of some of the mutations reported here on nucleation are also mediated by a change in the concentration of Aß42 rather than by an increase in a kinetic rate parameter. Some of the variants evaluated here may have additional effects, for example altering cleavage of APP. Future work will be needed to test these hypotheses.”

“Finally, the strikingly consistent effects of the dominant fAD mutations in our assay further strengthen the evidence that fAD is a “nucleation disease” ultimately caused by an increased rate of amyloid nucleation (Aprile et al., 2017; Cohen et al., 2018; Knowles et al., 2009). This accelerated nucleation can be caused by the direct effects of mutations – such as those quantified here – or by changes in upstream factors (Szaruga et al., 2017). If this hypothesis is correct, then nucleation is the key biochemical step to target to prevent or treat AD.”

2) Potential bias of the assay itself: coupling Aβ aggregation to sup35 aggregation and immobilizing the N-terminal part of Aβ in a linker (from reviewer 2):a) The specific purpose of this work is to map the genetic landscape for nucleation of Aβ42. Yet the reporter used here is the ability of a fragment of another amyloid (prion) protein sup35 to nucleate endogenous sup35. This is a totally different aggregation prone sequence than Aβ (Q/N rich instead of hydrophobic) with different intrinsic kinetics. What are the controls guaranteeing that a)Aβ42 is indeed rate limiting, b) that sup35 seeding is scaling along with Aβ42 nucleation (some Aβ mutations might lead to many small seeds while other to less but bigger seeds) and c) that both Aβ42 and sup35N domain aggregation are independent of each other in this fusion. How can antagonistic or synergistic effects be excluded?

We agree these are very important questions in relation to this specific assay. We state that Aβ is rate limiting in this assay because the nucleation domain of Sup35 (SupN) alone leads to no nucleation and no yeast growth in selective conditions (lacking adenine). These control experiments are presented in Chandramowlishwaran et al. (2018) and have been repeated by us for further validation (now included as new Figure 1—figure supplement 1A). Finally, the ability to grow without adenine depends on the recruitment and function of endogenous Sup35. Consistent with this, expression of Aβ42 alone also results in no detectable growth (Figure 1—figure supplement 1A).

Work from the Lindquist and Chernoff labs showed that amyloid sequences forming many unstable aggregates lead to more growth in the lack of adenine compared to sequences that instead could form highly stable amyloids (Frederick et al., 2014; Chandramowlishwaran et al., 2018). In line with this we observe one of the highest nucleation scores for a variant known to populate persistent oligomeric species, while very low nucleation scores for variants such as Aβ40, which are known to slowly nucleate long fibrils (Bolognesi, 2014, Sanagavarapu, 2019).

b) The sup35 fragment is N-terminally fused to Aβ42 and the data show biggest effects on the aggregation propensity at the C-terminus. Is the fact that the 16-23 fragment is less prominent not a bias resulting of transforming the N-terminal unstructured part of Aβ42 into a linker in the fusion thereby underestimating effect towards the N terminal part of the domain?

That mutations in the C-terminus more often reduce nucleation is highly expected given that the C-terminus forms the hydrophobic amyloid core in most of the published Aβ42 fibrillar structures. In addition, we would like to argue the following:

If the mutational effects we measure were biased by the N-terminal fusion, then one would expect a gradient of effects: small changes in NS close to the fusion and larger changes in NS further away from it. However the effect we see is instead modular and several mutations with large effects on nucleation also exist at the N-terminus (examples: H14I, Q15P, E22G)

Known fAD mutations are located mostly at the N-terminus, they all have significant effects on nucleation and are all correctly classified by this assay (Figure 2 and Figure 4)

We have tested whether fusing the Sup35 fragment next to the C-terminal core interferes with nucleation and it does not. Specifically, we quantified the nucleation of three C-terminal fragments of the peptide (aa 22-42, 24-42, 27-42) with Sup35 fused at their N-terminus and found that they nucleate similarly or better than full length Aß42. These data are included as Figure 3—figure supplement 1C and are reported in the main text.

c) The authors validate their assay by showing that previously measured nucleation rates of disease mutants correlate with their “nucleation” enrichment scores. The problem is that this only represents a handful of mutants (5 mutants) the majority of which are situated along positions 21/22/23 of the Aβ peptide. There is no guarantee that mutants in very different contexts such as in the flexible N-terminal part of the region or the C-terminal aggregation prone region will respond in such a nicely correlated manner.

The quantitative data (Yang et al., 2018) that we compare to consists of mutations at positions 21, 22 and 23. In addition to this quantitative data, we have collated the qualitative effects of 16 different mutations studied in vitro in ten previous publications (Supplementary file 1). Our quantitative data agrees with these previously reported effects of mutations in 14 out of 16 cases. These include mutations in the N-terminus, such as H6R and E11K. In two cases our data disagree with the literature:

D7H, an fAD variant, increases nucleation in our assay but showed a longer lag phase in the only in vitro kinetics assay that we could find in the literature (Chen et al., 2012).

A21G, another fAD variant, increases nucleation in our assay but has been reported to have various effects in vitro in different papers: it was reported to aggregate similarly to wild-type Aβ42 (Yang et al., 2018), or to have a decreased aggregation rate (Thu et al., 2019)

Our results suggest that D7H and A21G do, at least in certain conditions, increase nucleation like all the other dominant fAD mutations. In relation to this point, we have added an additional table (Supplementary file 1) and the following sentence to the text:

“Comparing our in vivo enrichment scores to the qualitative effects of 16 mutations analyzed in vitro across ten previous publications validated the assay, with mutational effects matching the effects on in vitro nucleation previously reported for 14 Aß variants out of 16. (Supplementary file 1).”

3) Overstatements in the Discussion that we hope can be addressed by making textual adaptations (all reviewers):a) The authors conclude that their results are more relevant to disease because they better report on nucleation than the study of Gray et al. They also conclude that they report on another mode of aggregation. I don't think there is evidence for that in this manuscript. First, although nucleation plays an important role in amyloid diseases this is not necessarily all of it. Second, the study of Gray et al. -although not correlating well with nucleation data- still identifies the importance of the 16-23 region and the C-terminal region in a more balanced manner. Both region are known to be crucial in determining Aβ nucleation.

We agree that the wording of the text may have been misleading in this sense and we have revised it accordingly to clarify that this refers specifically to the abilities of the assays to identify the known disease mutations. This is a factual statement – the scores from the current assay are a much better predictor of the known fAD mutations that the scores from the DHFR assay (area under the receiver operating curve, ROC-AUC=0.9 vs. AUC=0.5; AUC=1 is a perfect classifier, AUC=0.5 is a random classifier).

b) I find the Discussion to be unnecessarily dismissive of animal models and other model systems. Particularly "this simple system is now better-validated as a model of fAD than any other, including animal models where the effects of only one or a few mutations (including control mutations) have ever been tested.". The current paper models only one aspect of Aβ biology, aggregation, and doesn't take into account Aβ generation nor the mechanisms linking Aβ to neurodegeneration. Thus I think this required rewriting to acknowledge that each model has its strengths/limitations

We agree and have modified the text according to stress that we are specifically referring to the possibility of evaluating the “genetic agreement” between an assay and the clinical genetics of fAD. There are of course other measures by which one could quantify the consistency of an assay with a disease, such as phenotypic agreement. Our point, which we think is an important one, is that only by testing many disease-causing mutations and many random (or non-disease causing) mutations in an assay can one properly quantify how well the effects of mutations on the biochemical process(es) that it reports on match the effects of mutations in causing a human genetic disease. Based on our and other labs’ deep mutagenesis of multiple human disease proteins now, we think it can be dangerous to make strong conclusions about the validity of an assay by testing the behaviour of 1 or 2 disease-causing mutations and 1 or 2 control mutations. We have edited the Abstract of the manuscript and the Discussion section:

“The range of phenotypes that can be assessed and their similarity to the pathology of AD human brains are appealing features of many animal models of AD and many important insights have been derived – and will continue to be derived – from animal models (Sasaguri et al., 2017). However, there are applications where animal models cannot be realistically used, for example for high-throughput compound screening for drug discovery and for testing hundreds or thousands of genetic variants of unknown significance. For these applications, in vitro or cell-based (Pimenova and Goate, 2020; Veugelen et al., 2016) assays are required and an important challenge is to compare the “disease relevance” of different assays.”

c) Likewise there is an omission of other cell based and cell free models that have been used for assigning significance to pathological variants and should be incorporated into the Discussion, e.g. PMID: 32032730, PMID: 27100199

We agree and have cited these other assays in the Discussion.

d) The statement "Many in vitro and in vivo assays are proposed as “disease models” in biomedical research with their relevance often justified by how “physiological” the assays seem or how well phenotypes observed in the model match those observed in the human disease. However, such criteria are largely subjective, and assays that seem relevant to a disease may actually turn out to be reporting on irrelevant biochemical events, resulting in the of drugs that then fail in clinical trials." is vague – can the authors give specific examples and references with relevance to AD?

One could argue that the negative results from >400 clinical trials for AD support this statement. However, it is difficult to draw strong conclusions from negative results because of several confounders so we prefer not to highlight specific examples and have removed the second sentence quoted above. Our aim in this section is simply to raise a discussion point that more groups or companies should use deep mutational scanning to quantify how well there in vitro and cell-based assays “genetically agree” with specific human genetic diseases.

We believe that the other comments are straightforward to address with further changes to the text.Reviewer #1:[…]• The selection assay (reporting on nucleation) seems to be highly reproducible, as supported by data presented in Figure 1B. Nevertheless, this reviewer wonders about the potential impact of the Sup35N fusion on Aβ solubility/hydrophobicity and aggregation. In addition, Aβ aggregation has been reported to be concentration dependent and it is not clear from the data whether the mutant peptides are, or not, expressed at similar levels.Could peptide fusion and/or expression levels have changed the intrinsic attributes of Aβ42 and could these potential alterations explain -at least to some degree- the mismatch observed between nucleation and hydrophobicity (Figure 1D, 3F)?

We agree with the reviewer that this is a limitation of our assay and at the moment cannot exclude that the possible variability in expression levels among different Aβ42 variants may have an influence on the nucleation scores that we quantify. This is a limitation of most deep mutation scanning assays, as it is particularly challenging to accurately track expression levels of thousands of protein variants in parallel. This was achieved in some cases by fluorescent-tagging the mutagenized protein and analyzing the library by flow cytometry (Staller et al., 2017), a solution which is unfortunately not possible to adopt in the conditions of our assay. We have added the following sentence to acknowledge this limitation.

“It is possible that the effects of some of the mutations reported here on nucleation are also mediated by a change in the concentration of Aß42 rather than by an increase in a kinetic rate parameter. In addition, some of the variants evaluated here may have additional effects, for example altering cleavage of APP. Future work will be needed to test these hypotheses.”

• Figure 2 presents the effects of the amino acid changes on the nucleation of Aβ. The authors mention that all possible amino acid changes in Aβ42 were introduced; however, there are several white spots corresponding to non-existing mutants. Why these mutants are missing?

The sentence refers to nt substitutions, rather than codon substitutions. Indeed, our library contains all the possible 378 nt changes to the Aβ sequence. In order to build the Aβ mutant library we used an error-prone PCR approach which introduces changes at the nucleotide level, thus limiting the complete coverage of all codon substitutions. However, we optimised the approach to ensure the coverage of all single nt substitutions and to maximise the number of double nt substitutions (47.2%). The “white spots” in Figure 2 therefore represents codons substitutions not represented in our library.

• The authors proposed and discussed the presence of gatekeepers of nucleation. How precisely these have been selected? Is there any cut-off score? Data in Figure 2D does not clarify this point: data for H13 or Q15 do not seem to be much different from the positions selected. Furthermore, only few mutations seem to have been tested at position A42 and an equal number of mutations increase or decrease nucleation, is this residue/position truly a gatekeeper?

Gatekeeper positions were defined as those where as many (or more) single aa substitutions increase as decrease nucleation for FDR=0.1. For the definition of gatekeepers, we only considered those positions where most mutations are significantly different from WT (FDR=0.1), i.e.: If the majority of mutations at a specific position are WT-like then this position is not considered a gate-keeper according to our definition. In this line, we defined position 42 as a gatekeeper. This is now directly stated in the Materials and methods.

In this revised version of the manuscript we have also updated Figure 2D and Figure 2—figure supplement 1A and B because we found a minor error in the script that generated the plot in the previous version. As can be easily seen in the new figures, this change does not affect any of our results or conclusions.

• In the last part of the manuscript, the authors analysed the nucleation of fAD-linked mutations in Aβ. The figures display "ROC curves built using 12 fAD mutants versus all other single aa mutants in the 484 datasets for variant effect and aggregation predictors". This reviewer would appreciate a figure where data for each specific fAD variant is displayed; or alternatively a table presenting the key data from a case by case analysis. It would be also interesting to see how the data relates to clinical onset.

Data for each fAD variant is now included in Supplementary file 1. We also included a new figure reporting on the relation between nucleation scores and clinical onset age for a subset of six mutations (Figure 4—figure supplement 1E). However, the age-of-onset data is too noisy for us to draw any conclusion.

• The conclusions that "the rate of nucleation in a cell-based assay accurately identifies mutations that cause dominant familial Alzheimer's disease" and "AD is actually a nucleation disease" should be reviewed, while considering that several other “very aggressive” mutations in APP and in Presenilins (γ secretase) do not generated mutant, but wild type Aβ peptides. In fact, the most aggressive mutations in APP (such as the Iberian or Autrian) and mutations in Presenilin share a common mechanism that results in the enhanced generation of longer Aβ peptides (Szaruga et al., 2017). How do the authors incorporate these findings into their model?In addition, it is known that pathogenic mutations in Aβ1-42 do not only affect its aggregation propensity but also the cleavage of APP by β-secretase. The protective, Icelandic APP variant is one of these cases.In short, the Discussion would benefit from a broader perspective.

We agree and have expanded the Introduction and the discussion of our results. The importance of accelerated nucleation as the cause of fAD is actually also in line with the proposed mechanism explaining the effect of fAD mutations beyond Aβ, such as the ones in PSEN1, PSEN2, and in APP outside of the Aβ region. A common effect of these mutations is to enhance the generation of Aβ42 over shorter versions of the peptide (Szaruga et al., 2017), therefore increasing Aβ42 relative and/or absolute concentration and facilitating nucleation. In addition, the substrate-enzyme destabilization caused by some mutations in PSEN1 and PSEN2 leads to an increase in the representation of longer Aβ peptides (≥Aβ42) displaying increased nucleation propensity and increased neurotoxicity (Veugelen et al., 2016; Benitova et al., 2012; Vandersteen et al., 2012; Conicella et al., 2014). We cannot exclude that some of the mutations in our library, especially those in the very first or last residues of the peptide, could impact cleavage of APP in humans and lead to over-representation of longer Aβ peptides.

• The authors should note that the pathogeneicity of the H6R is unclear.

A comment on H6R has been added.

Reviewer #2:[…]More specifically:1) The specific purpose of this work is to map the genetic landscape for nucleation of Aβ42. Yet the reporter used here is the ability of a fragment of another amyloid (prion) protein sup35 to nucleate endogenous sup35. This is a totally different aggregation prone sequence than Aβ (Q/N rich instead of hydrophobic) with different intrinsic kinetics. What are the controls guaranteeing that a)Aβ42 is indeed rate limiting, b) that sup35 seeding is scaling along with Aβ42 nucleation (some Aβ mutations might lead to many small seeds while other to less but bigger seeds) and c) that both Aβ42 and sup35N domain aggregation are independent of each other in this fusion. How can antagonistic or synergistic effects be excluded?

We agree these are very important questions in relation to this specific assay. We state that Aβ is rate limiting in this assay because the nucleation domain of Sup35 (Sup35N) alone leads to no nucleation and no yeast growth in selective conditions (lacking adenine). These crucial control experiments are presented in Chandramowlishwaran et al. (2018) and have been repeated by us for further validation (now included as Figure 1—figure supplement 1A). Finally, the ability to grow without adenine depends on the recruitment and function of endogenous Sup35. According to this, expression of Aβ42 alone also results in no detectable growth (Figure 1—figure supplement 1A).

The Lindquist and Chernoff labs showed that amyloid sequences forming many unstable aggregates lead to more growth in the lack of adenine compared to sequences that instead could form highly stable amyloids (Frederick et al., 2014; Chandramowlishwaran et al., 2018). In line with this we observe one of the highest nucleation scores for a variant known to populate persistent oligomeric species, while very low nucleation scores for variants such as Aβ40, which are known to slowly nucleate long fibrils (Bolognesi, 2014).

2) The sup35 fragment is N-terminally fused to Aβ42 and the data show biggest effects on the aggregation propensity at the C-terminus. Is the fact that the 16-23 fragment is less prominent not a bias resulting of transforming the N-terminal unstructured part of Aβ42 into a linker in the fusion thereby underestimating effect towards the N terminal part of the domain?

That mutations in the C-terminus more often reduce nucleation is highly expected given that the C-terminus forms the hydrophobic amyloid core in most of the published Aβ42 fibrillar structures. In addition, we would like to argue the following:

If the mutational effects we measure were biased by the N-terminal fusion, then one would expect a gradient of effects: small changes in NS close to the fusion and larger changes in NS further away from it. However the effect we see is instead modular and several mutations with large effects on nucleation also exist at the N-terminus (examples: H14I, Q15P, E22G)

Known fAD mutations are located mostly at the N-terminus, they all have significant effects on nucleation and are all correctly classified by this assay (Figure 2 and Figure 4)

We have tested whether fusing the Sup35 fragment next to the C-terminal core interferes with nucleation and it does not. Specifically, we quantified the nucleation of three C-terminal fragments of the peptide (aa 22-42, 24-42, 27-42) with Sup35 fused at their N-terminus and found that they nucleate similarly or better than full length Aß42. These data are included as Figure 3—figure supplement 1C and are reported in the main text.

3) The authors validate their assay by showing that previously measured nucleation rates of disease mutants correlate with their “nucleation” enrichment scores. The problem is that this only represents a handful of mutants (5 mutants) the majority of which are situated along positions 21/22/23 of the Aβ peptide. There is no guarantee that mutants in very different contexts such as in the flexible N-terminal part of the region or the C-terminal aggregation prone region will respond in such a nicely correlated manner.Overall therefore the absence of bias in the current experimental setup needs to be addressed by a more rigorous validation. Mutants that increase/decrease nucleation along different parts of the sequence should be experimentally validated in the same manner than the Yang et al., 2018 paper currently used to show correlation.Finally, the authors conclude that their results are more relevant to disease because they better report on nucleation than the study of Gray et al. They also conclude that they report on another mode of aggregation. I don't think there is evidence for that in this manuscript. First, although nucleation plays an important role in amyloid diseases this is not necessarily all of it. Second, the study of Gray et al. -although not correlating well with nucleation data- still identifies the importance of the 16-23 region and the C-terminal region in a more balanced manner. Both region are known to be crucial in determining Aβ nucleation.

The quantitative data (Yang et al., 2018) that we compare to consists of mutations at positions 21, 22, 23.

In addition to this quantitative data, the qualitative effects of 16 different mutations have previously been analyzed in vitro across ten previous publications (Supplementary file 1). Our quantitative data agrees with these previously reported effects of mutations in 14 out of 16 cases. These include mutations in the N-terminus, such as H6R and E11K. In two cases our data disagree with the literature:

D7H, an fAD variant, increases nucleation in our assay but showed a longer lag phase in the only in vitro kinetics assay that we could find in the literature (Chen et al., 2012).

A21G, another fAD variant, increases nucleation in our assay but has been reported to have various effects in vitro in different papers: it was reported to aggregate similarly to wild-type Aβ42 (Yang et al., 2018), to have a decreased aggregation rate (Thu et al., 2019), and appears to accelerate aggregation in Vandersteen et al., 2012. Our results suggest that A21G does, at least in certain conditions, increase nucleation like all the other dominant fAD mutations in our assay.

In relation to this point, we have added an additional table (Supplementary file 1) and the following sentence to the text:

“Comparing our in vivo enrichment scores to the qualitative effects of 16 mutations analyzed in vitro across ten previous publications validated the assay, with mutational effects matching the effects on in vitro nucleation previously reported for 14 Aß variants out of 16. (Supplementary file 1).”

Finally, there are some discrepancies in the mutation effects reported in the literature for A2V. Recent kinetics analysis reports a decrease both in primary and secondary nucleation for this variant (Meisl et al., 2016, Murray et al., 2016), in line with our results. However, previous work found the overall nucleation rate for A2V to be similar to wild-type Aβ42 (Benilova et al., 2014). Interestingly, the same study shows an increase in nucleation for A2V Aβ40, compared to wild-type Aβ40. This difference in mutation effect actually encourages the use of a deep mutagenesis approach for the study of mutation effects on Aβ40. We should add that A2V is a recessive AD mutation, suggesting that the mechanism by which it leads to toxicity may differ from that underlying dominant fAD mutations and, for example, may involve modulation of B-cleavage efficiency with increased Aβ production, as suggested by Benilova *et al.* A sentence on this has been added to the main text.

Reviewer #3:In this manuscript, the authors use an in vitro nucleation assay to assess the aggregation properties of 14 483 mutated forms of Aβ, encompassing all possible single amino acid changes as well as double amino acid mutations. They identify mutations that both enhance and suppress Aβ nucleation, and define modular regions of Aβ linked to various aggregation properties. Importantly, the assay is able to identify known fAD mutations. Based on these results, the authors conclude that their system provides a rapid, cost effective means to assign pathogenicity to Aβ variants of unknown significance.Although I find the work to be technically sound, and to provide interesting insights into the biochemical properties of Aβ, I think the conclusions, particular with regards to disease relevance, to be somewhat overstated in part and there are some omissions of relevant literature. Some specific points are listed below:1) The authors state that "Moreover, given the human mutation rate and population size, it is likely that nearly all of these possible variants in Aβ actually exist in at least one individual currently alive on the planet" – however, these mutations may not be compatible with life, and as the majority (14015) examined are double mutations, how likely are these to exist in people? I am not sure that the species with double amino acid alterations bear relevance to disease or are likely to exist in individuals

The comment refers to all possible single nucleotide changes. Given the human germline mutation rate (~5*10^-8^) (Scally et al., 2016) and the number of individuals currently alive (~8*10^9^), all of the Aβ42 variants that are compatible with life are likely to currently exist in at least one individual currently alive on the planet simply from de novo mutations each generation.

We included double nt changes for two reasons: (1) to increase the number of amino acid substitutions and (2) to include double amino acid changes to provide larger changes in physico-chemical parameters. The advantage of this can be seen for example in the analysis of the effects of charge where the double mutants allow us to quantify the effects of varying both the net charge and total charge (Figure 3).

2) The manuscript states that it is not clear why APP/Aβ mutations cause AD, but it would be good to see a discussion of recent mechanistic work demonstrating that fAD mutations destabilise APP-GSEC complexes, resulting in the release of longer Aβ peptides (PMID: 28753424). Thus, the mutations could be shifting the profile of Aβ species produced as well as influencing aggregation, this should be discussed.

We have added the following sentences to the Discussion:

“That accelerated nucleation is a common cause of fAD is also supported by the effects of mutations in *APP* outside of Aß42 and by the effects of mutations in *PSEN1* and *PSEN2*. These mutations destabilize enzyme-substrate complexes, increasing the production of the longer Aß42 peptide that more effectively nucleates amyloid formation (Szaruga et al., 2017; Veugelen et al., 2016). In addition, Aß42 oligomers are hypothesized to be more toxic (Michaels et al., 2020; Bolognesi et al., 2010). It is possible that the effects of some of the mutations reported here on nucleation are also mediated by a change in the concentration of Aß42 rather than by an increase in a kinetic rate parameter. Some of the variants evaluated here may have additional effects, for example altering cleavage of APP. Future work will be needed to test these hypotheses.”

3) Relating to point 2, I think the Introduction should include a description of the tripeptide cleavage pathways that result in multiple forms of Aβ, and also recent work that Aβ 43 is more neurotoxic and aggregation prone than 42.

We have added the following sentences to the Introduction, in addition to the extended text in the Discussion reported in point 2:

“Several mutations in *PSEN1* and *PSEN2*, the genes coding for the secretases performing sequential cleavage of APP, also lead to autosomal dominant forms of AD.”

4) I find the Discussion to be unnecessarily dismissive of animal models and other model systems. Particularly "this simple system is now better-validated as a model of fAD than any other, including animal models where the effects of only one or a few mutations (including control mutations) have ever been tested.". The current paper models only one aspect of Aβ biology, aggregation, and doesn't take into account Aβ generation nor the mechanisms linking Aβ to neurodegeneration. Thus I think this required rewriting to acknowledge that each model has its strengths/limitations

As stated above, we have rephrased the Discussion to tone down some of these statements and to correctly acknowledge the importance of other models. Apologies for this.

5) Likewise there is an omission of other cell based and cell free models that have been used for assigning significance to pathological variants and should be incorporated into the Discussion, e.g. PMID: 32032730, PMID: 27100199

We have rephrased the Discussion to tone down some of these statements and to acknowledge the relevance of other models and assays, such as these ones, which are now cited in the main text.

6) The statement "Many in vitro and in vivo assays are proposed as 'disease models' in biomedical research with their relevance often justified by how 'physiological' the assays seem or how well phenotypes observed in the model match those observed in the human disease. However, such criteria are largely subjective, and assays that seem relevant to a disease may actually turn out to be reporting on irrelevant biochemical events, resulting in the of drugs that then fail in clinical trials." is vague – can the authors give specific examples and references with relevance to AD?

One could argue that the negative results from >400 clinical trials for AD support this statement. However, it is difficult to draw strong conclusions from negative results because of other potential causes for trial failure, so we prefer not to highlight specific examples and have deleted the second sentence quoted above. Our goal in this section is simply to raise the discussion point of encouraging more groups to use deep mutational scanning to quantify how well their assays ‘genetically agree’ with specific human genetic diseases.